# Associative learning drives longitudinally graded presynaptic plasticity of neurotransmitter release along axonal compartments

Aaron Stahl[1], Nathaniel C Noyes[1], Tamara Boto[1], Valentina Botero[1], Connor N Broyles[1], Miao Jing[2], Jianzhi Zeng[3,4,5], Lanikea B King[1], Yulong Li[2,3,4,5], Ronald L Davis[1], Seth M Tomchik[1]*

[1]Department of Neuroscience, The Scripps Research Institute, Jupiter, United States; [2]Chinese Institute for Brain Research, Beijing, China; [3]Peking-Tsinghua Center for Life Sciences, Peking University, Beijing, China; [4]State Key Laboratory of Membrane Biology, Peking University School of Life Sciences, Beijing, China; [5]PKU IDG/McGovern Institute for Brain Research, Beijing, China

**\*For correspondence:** STomchik@scripps.edu

**Competing interest:** The authors declare that no competing interests exist.

**Abstract** Anatomical and physiological compartmentalization of neurons is a mechanism to increase the computational capacity of a circuit, and a major question is what role axonal compartmentalization plays. Axonal compartmentalization may enable localized, presynaptic plasticity to alter neuronal output in a flexible, experience-dependent manner. Here, we show that olfactory learning generates compartmentalized, bidirectional plasticity of acetylcholine release that varies across the longitudinal compartments of *Drosophila* mushroom body (MB) axons. The directionality of the learning-induced plasticity depends on the valence of the learning event (aversive vs. appetitive), varies linearly across proximal to distal compartments following appetitive conditioning, and correlates with learning-induced changes in downstream mushroom body output neurons (MBONs) that modulate behavioral action selection. Potentiation of acetylcholine release was dependent on the Ca$_V$2.1 calcium channel subunit *cacophony*. In addition, contrast between the positive conditioned stimulus and other odors required the inositol triphosphate receptor, which maintained responsivity to odors upon repeated presentations, preventing adaptation. Downstream from the MB, a set of MBONs that receive their input from the γ3 MB compartment were required for normal appetitive learning, suggesting that they represent a key node through which reward learning influences decision-making. These data demonstrate that learning drives valence-correlated, compartmentalized, bidirectional potentiation, and depression of synaptic neurotransmitter release, which rely on distinct mechanisms and are distributed across axonal compartments in a learning circuit.

## Editor's evaluation

This manuscript will be of interest to scientists working on learning and memory and synaptic plasticity. The study mostly uses an acetylcholine sensor in the fly brain to image activity, which is novel and helps to tie together previous studies reporting memory-induced changes in calcium transients. In particular, the study highlights the compartmentalised plasticity along Kenyon cell axon terminals, the main cell type of the insect mushroom body.

## Introduction

Neuronal dendrites carry out computations through compartmentalized signaling, while axons have long been considered to carry signals to their terminal fields relatively uniformly following spike initiation. However, anatomical and physiological compartmentalization of axons has been recently documented in neurons from worms through mammals (*Boto et al., 2014*; *Cohn et al., 2015*; *Hendricks et al., 2012*; *Rowan et al., 2016*). How axonal compartmentalization influences information flow across neuronal circuits and modulates behavioral outcomes is not understood. One functional role for axonal compartmentalization may be to enable localized, presynaptic plasticity to alter output from select axon compartments in a flexible, experience-dependent manner. This would vastly enhance the neuron's flexibility and computational capabilities. A potential function of such compartmentalization would allow independent modulation of axonal segments and/or synaptic release sites by biologically salient events, such as sensory stimuli that drive learning.

The anatomical organization of the *Drosophila* mushroom body (MB) makes it an exemplary test bed to study how sensory information is processed during learning and rerouted to alter behavioral outcomes. The MB encodes odor in sparse representations across the intrinsic MB neurons, Kenyon cells (KCs), which are arranged in several parallel sets. They project axons in fasciculated bundles into several anatomically distinct, but spatially adjacent lobes (α/β, α′/β′, and γ) (*Crittenden et al., 1998*). KC axons are longitudinally subdivided into discrete tiled compartments (*Aso et al., 2014a*). Each compartment receives afferent neuromodulatory input from unique dopaminergic neurons (*Aso et al., 2014a*; *Mao and Davis, 2009*), and innervates unique efferent mushroom body output neurons (MBONs) (*Aso et al., 2014a*). Each set of dopaminergic neurons plays an individual role in learning, with some modulating aversive learning (*Mao and Davis, 2009*; *Schroll et al., 2006*; *Schwaerzel et al., 2003*), others modulating reward learning (*Liu et al., 2012*; *Yamagata et al., 2015*), and a third class modulating memory strength without driving behavioral valence (*Boto et al., 2019*). Similarly, each MBON has a unique effect on behavioral approach and avoidance, with some biasing the animal to approach, others biasing the animal to avoidance, and some having no known effect (*Aso et al., 2014b*; *Perisse et al., 2016*; *Plaçais et al., 2013*; *Séjourné et al., 2011*).

A major question in learning and memory is how presynaptic plasticity contributes to reweighting the flow of sensory signals across each of the downstream 'approach' or 'avoidance' circuits, altering action selection and memory retrieval. In naïve conditions, *Drosophila* dopaminergic circuits modulate cAMP in a compartmentalized fashion along the MB axons (*Boto et al., 2014*). This compartmentalized dopaminergic signaling can independently modulate $Ca^{2+}$ responses in each compartment, as well as the responses of the downstream valence-coding MBONs (*Cohn et al., 2015*). Presynaptic plasticity within each KC compartment likely contributes to the changes in downstream MBON activity that guide learned behavioral responses (*Zhang et al., 2019*). However, manipulation of the 'aversive' protocerebral posterior lateral 1 (PPL1) dopaminergic neurons does not detectably alter $Ca^{2+}$ signals in KCs (*Boto et al., 2019*; *Hige et al., 2015a*). Furthermore, $Ca^{2+}$ responses in KCs are uniformly potentiated across compartments with appetitive classical conditioning protocols and unaltered in KCs following aversive protocols (*Louis et al., 2018*). This raises the question of how local, compartmentalized synaptic plasticity in KCs drives coherent changes across the array of downstream MBONs to modulate action selection during memory retrieval. Learning/dopamine-induced plasticity has been demonstrated in the MBONs (*Berry et al., 2018*; *Hige et al., 2015a*; *Hige et al., 2015b*; *Owald et al., 2015*), with dopamine also acting directly on them (*Takemura et al., 2017*) (in addition to KCs). Feedforward inhibition among MBONs that drive opposing behavioral outcomes provides a mechanism explaining how bidirectional valence coding in MBONs could be generated, with or without bidirectional presynaptic plasticity (*Perisse et al., 2016*). The compartmentalized, dopamine-dependent plasticity in KCs and the necessity for dopamine receptors and downstream signaling molecules in the intrinsic KCs points to a potential presynaptic contribution (*Kim et al., 2007*; *McGuire et al., 2003*; *Zars et al., 2000*). Thus, compartmentalized presynaptic plasticity could contribute to reweighting the flow of olfactory information to downstream circuits.

Here, we describe how learning alters the flow of information through the MB via alteration of synaptic release of the KC neurotransmitter acetylcholine (ACh) (*Barnstedt et al., 2016*), using a genetically encoded indicator of ACh neurotransmission. The data reveal that appetitive and aversive learning alter compartmentalized acetylcholine release in distinct spatial patterns, with differing

molecular mechanisms, coherently reweighting the flow of olfactory information across the ensemble of downstream neurons that mediate action selection.

## Results

### Associative learning modulates neurotransmitter release in a spatially distinct manner across longitudinal axonal compartments

Synapses within each MB compartment transmit olfactory information from KCs to compartment-specific MBONs (*Figure 1*, Figure 5 A, Figure 5-figuresupplement 1A ; *Aso et al., 2014a*; *Tanaka et al., 2008*). The MBONs exert distinct and often-opposing effects on behavior, with some innately promoting approach and others promoting avoidance (*Aso et al., 2014b*; *Berry et al., 2018*; *Ichinose et al., 2015*; *Owald et al., 2015*; *Perisse et al., 2016*; *Plaçais et al., 2013*; *Sayin et al., 2019*; *Séjourné et al., 2011*). Synaptic depression has been observed (postsynaptically) at KC-MBON synapses following pairing of odor with stimulation of PPL1 neurons that are critical for aversive learning (*Hige et al., 2015a*), suggesting that depression could be the primary learning rule implemented at these synapses. While some MBONs exhibit bidirectional responses to conditioning, the major described mechanism involves a sign change that occurs postsynaptic to the KCs (polysynaptic feedforward inhibition) (*Owald et al., 2015*; *Perisse et al., 2016*) and the presynaptic contributions remain unknown. To test for the presence, directionality, and variation of presynaptic plasticity across MB axonal compartments, we expressed a synaptic ACh sensor to monitor neurotransmitter release from KCs in vivo (*Zhang et al., 2019*). The genetically encoded ACh reporter, GPCR-Activation–Based-ACh sensor (GRAB-ACh) (*Figure 1A*; *Jing et al., 2020*; *Jing et al., 2018*; *Zhang et al., 2019*), was expressed in KCs using the broad KC driver 238Y-Gal4. Appetitive conditioning was carried out, monitoring ACh release from the γ lobe compartments evoked by the olfactory conditioned stimuli (CS+ and CS-) before and after pairing the CS+ with a sucrose unconditioned stimulus (US) (*Figure 1—figure supplement 1*). Responses were compared to those in odor-only control cohorts to determine whether any learning-induced changes resulted from potentiation or depression. We quantified several parameters (*Figure 1—figure supplement 1*), starting with how the response to each odor changed after conditioning (e.g. *Figure 1D and F*). This was collapsed to a single value, the within-treatment Δ(post/pre), for comparison across conditions. The Δ(post/pre) of the CS+ and CS-was compared to determine how each changed relative to the other, and then each was compared to its respective odor-only control to quantify whether it was potentiated or depressed, accounting for any non-associative olfactory adaptation (*Figure 1G-K*, *Figure 1—figure supplement 2*).

Appetitive conditioning produced plasticity in ACh release that varied across the axonal compartments of the MB γ lobe in several key ways (*Figure 1*). Conditioning significantly increased CS+ responses relative to the CS- responses in the three most proximal γ lobe compartments: γ1, γ2, and γ3 (*Figure 1C, D–*; *Figure 1—figure supplement 2*). In each of these compartments, this was due to different underlying dynamics. In the γ1 compartment, the CS+response was potentiated: following appetitive conditioning, the Δ(post/pre) CS+ response was significantly larger than the respective odor-only control (ethyl butyrate: EB), while the CS- did not differ from its odor-only control (isoamyl acetate: IA) (*Figure 1G*, *Figure 1—figure supplement 2*). In γ1, the CS+ response significantly increased following conditioning (*Figure 1D*). While the CS- response decreased following conditioning (*Figure 1D*), this decrease was indistinguishable from the rate of adaptation among odor-only controls in that compartment (*Figure 1G*). This suggests that the main effect in the γ1 compartment was potentiation of the CS+ response, and highlights the value of normalizing for non-associative olfactory adaptation. In the γ2 compartment, both the CS+ was potentiated and the CS- depressed relative to the odor-only controls (*Figure 1H*, *Figure 1—figure supplement 2*). In the γ3 compartment, while the CS+ and CS- differed, neither was significantly altered relative to odor-only controls. However, there was a trend toward depression in the CS- group (*P* = 0.099) (*Figure 1I*), as well as a significant decrease in the post-conditioning CS- response (*Figure 1—figure supplement 2*). Overall, these data reveal a spatial gradient of relative CS+ enhancement in the proximal γ compartments, shifting from CS+ potentiation in γ1 toward CS- depression in γ3, with the spatially intermediate γ2 exhibiting both (*Figure 1L*). This gradient of CS+:CS- plasticity suggests that both the CS+ and CS-contribute to learning by modulating compartmentalized KC (and downstream MBON) output, with different forms of plasticity in each compartment.

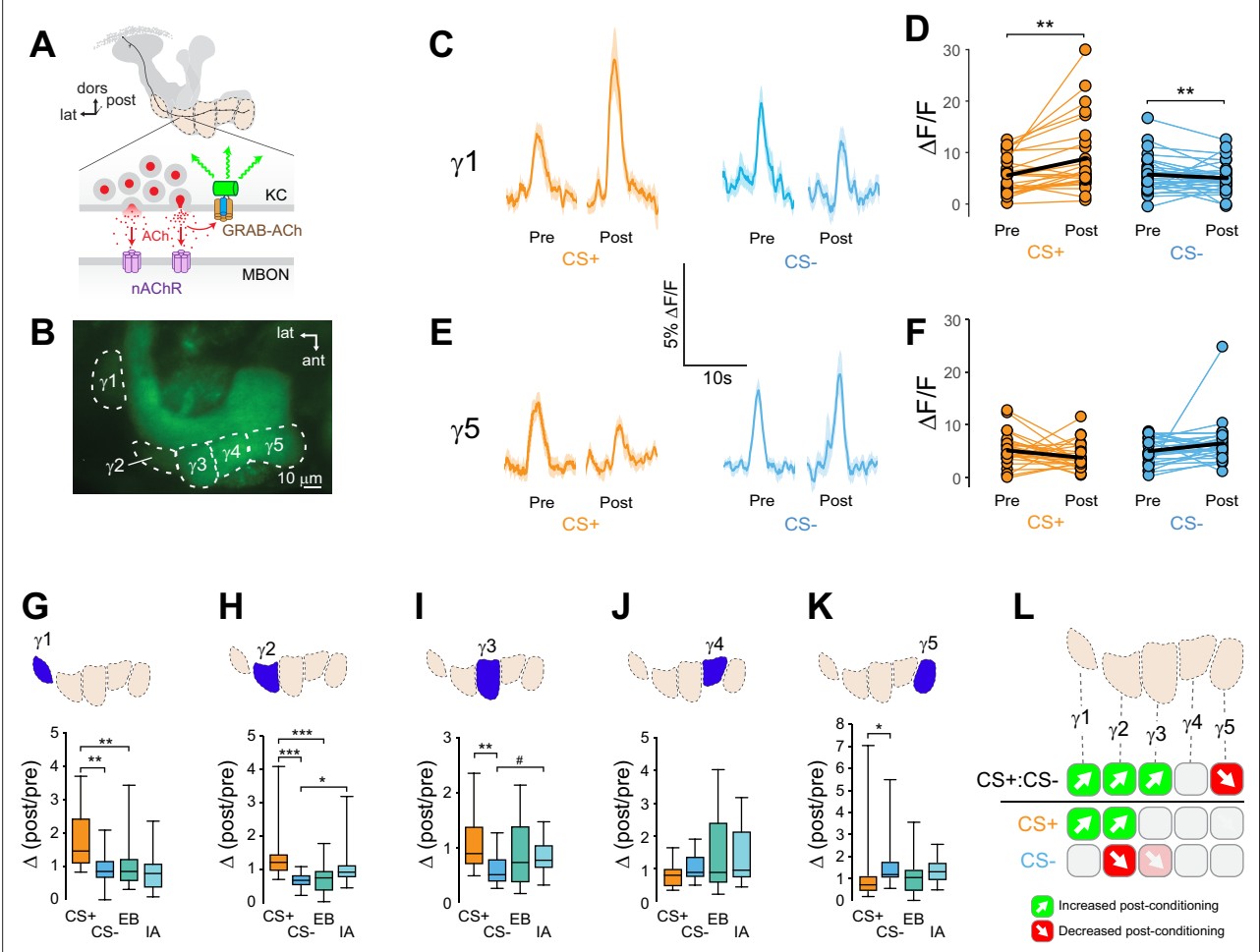

**Figure 1.** Compartment-specific alterations of acetylcholine (ACh) release in the mushroom body (MB) following appetitive conditioning. (**A**) Diagram of the GRAB-ACh reporter expressed in presynaptic terminals of a Kenyon cell (KC), viewed from a frontal plane. nAChR: nicotinic ACh receptor; dors: dorsal; lat: lateral, post: posterior; MBON: mushroom body output neuron. (**B**) Confocal image of GRAB-ACh driven in KCs with the 238Y-Gal4 driver. ant: anterior. (**C**) Time series traces of odor-evoked GRAB-ACh responses in the γ1 compartment, pre- and post-conditioning, for the CS+ (ethyl butyrate: EB) and CS- (isoamyl acetate: IA) odor. The line and shading represent the mean ± SEM. (**D**) Quantification of the pre- and post-conditioning responses to the CS+ and CS- from the γ1 compartment from individual animals, with the mean graphed as a black line. **p<0.01; n = 27 (Wilcoxon rank-sum test). (**E**) Time series traces imaged from the γ5 compartment, graphed as in panel C. (**F**) Quantification of peak responses from the γ5 compartment, graphed as in panel D. (**G–K**) Change in odor-evoked ACh release (Δ(post/pre) responses) following conditioning for the CS+, CS-, and odor-only controls (EB and IA). *p<0.05, **p <0.01, ***p<0.001; n = 27 (Kruskal–Wallis/Dunn). (**G**) γ1 compartment. (**H**) γ2 compartment. (**I**) γ3 compartment. #p = 0.099. (**J**) γ4 compartment. (**K**) γ5 compartment. (**L**) Summary of plasticity in ACh release across γ lobe compartments. Green up arrows indicate increases in the CS+:CS- (first row) or potentiation of the CS+ response relative to odor-only controls (second row), while red down arrows indicate decreases in the CS+:CS- (first row) or depression of the CS- relative to odor-only controls (third row).

The online version of this article includes the following figure supplement(s) for figure 1:

**Figure supplement 1.** The imaging protocol for conditioning and data analysis.

**Figure supplement 2.** Effects of appetitive conditioning on GRAB-ACh responses across the γ lobe compartments.

Moving further down the distal length of the KC axons, the plasticity produced by appetitive conditioning shifted from relative CS+ potentiation (↑CS+:CS-) to depression. In the most distal γ5 compartment, the CS+ response was reduced relative to the CS- following appetitive conditioning (***Figure 1E, F, K***, ***Figure 1—figure supplement 2***). The effect could not be unambiguously assigned to CS+ depression, though there was no evidence of CS- potentiation (***Figure 1K***, ***Figure 1—figure supplement 2***). In addition, control flies for RNAi experiments (discussed further below) exhibited a trend toward a reduction in the CS+:CS- response in the γ5 compartment following appetitive conditioning (Figure 3), accompanied by a significant decrease in CS+ response (***Figure 3—figure***

*supplement 1D*). These lines of evidence point to a depression in CS+ evoked ACh release from the distal γ5 compartment. Overall, appetitive conditioning increased relative CS+responsivity in γ1–γ3 compartments, which was derived from a proximal-to-distal gradient of CS+ potentiation to CS-depression, and reduction of relative CS+ responsivity in the γ5 compartment (*Figure 1L*). Thus, the plasticity was bidirectional between the proximal and distal axonal compartments. This likely contributes to approach behavior by simultaneously enhancing the conditioned odor-evoked activation of downstream 'approach' circuits and inhibiting 'avoidance' MBON circuits.

## Aversive conditioning drives ACh release in the opposite direction across MB compartments

The above data suggested that appetitive conditioning potentiated ACh synaptic release from the proximal γ lobe compartments. Yet synaptic depression is the main described plasticity mechanism at the KC-MBON synapses following olfactory conditioning (*Barnstedt et al., 2016*; *Modi et al., 2020*; *Owald et al., 2015*; *Perisse et al., 2016*; *Séjourné et al., 2011*; *Zhang and Roman, 2013*; *Zhang et al., 2019*). In the γ1 compartment, where it has been examined in detail with electrophysiology, aversive reinforcement via dopaminergic neuron stimulation produces synaptic depression (*Hige et al., 2015a*). Since many of the prior studies involved aversive reinforcement, we reasoned that appetitive and aversive conditioning may produce bidirectional plasticity of ACh release within each compartment, with the sign/directionality matching the postsynaptic MBON valence. To test this, we examined whether aversive conditioning produced the opposite effect in the same compartments as appetitive conditioning did. ACh release from KCs was imaged with GRAB-ACh and flies were trained with an aversive odor-shock conditioning protocol (*Figure 2A and B*). In these experiments, we focused on the γ2–γ5 compartments, as the fly was mounted at a higher angle, making the GRAB-ACh signal difficult to simultaneously visualize from γ1 along with that of the other compartments. Following aversive conditioning, there was a reduction in the CS+ response relative to the CS- in the γ2 and γ3 compartments (*Figure 2C-H*, *Figure 2—figure supplement 1*). This was due to depression in the CS+ response, as the Δ(post/pre) of the CS+ was significantly lower than the odor-only controls in each compartment (*Figure 2G and H*). The γ4 and γ5 compartments exhibited no significant change in ACh release (*Figure 2I, J* and *Figure 2—figure supplement 1*). When compared to appetitive conditioning, aversive conditioning produced plasticity in the opposite direction in the γ2 and γ3 compartments (*Figures 1L and 2K*). Thus, appetitive and aversive conditioning produced bidirectional plasticity within multiple compartments, which likely represents a presynaptic contribution to learning-induced changes in odor responsivity among postsynaptic MBONs following conditioning (*Berry et al., 2018*; *Hige et al., 2015a*; *Owald et al., 2015*; *Zhang et al., 2019*).

## Presynaptic potentiation relies on the *acophony* Ca$_V$2.1 Ca$^{2+}$ channel

Associative learning alters Ca$^{2+}$ transients in MB γ neurons (*Louis et al., 2018*), which could influence neurotransmitter release. Major sources of stimulus-evoked intracellular Ca$^{2+}$ include influx through voltage-sensitive Ca$_V$2 channels, which are involved in presynaptic short-term and homeostatic plasticity (*Frank et al., 2006*; *Inchauspe et al., 2004*; *Ishikawa et al., 2005*; *Müller and Davis, 2012*). To probe the Ca$^{2+}$-dependent molecular mechanisms underlying presynaptic plasticity, we first knocked down the α subunit of the Ca$_V$2 Ca$^{2+}$ channel encoded by *cacophony* (Cac). Cac was knocked down conditionally in adult MBs with RNAi using the strong, KC-selective R13F02-Gal4 driver, combined with the ubiquitous temperature-sensitive tub-Gal80$^{ts}$ repressor (*McGuire et al., 2003*) to reduce the potential for developmental effects (*Figure 3A*). The RNAi line was selected to moderately reduce, but not eliminate, Cac expression (*Brusich et al., 2015*). Quantitative reverse transcription polymerase chain reaction (RT-qPCR) analysis of Cac knockdown (driven ubiquitously with tubulin-Gal4) showed that conditional expression with this induction protocol reduced Cac levels ~29% (*Figure 3—figure supplement 1C*). For imaging experiments, RNAi expression was induced in the MB with R13F02-Gal4 following eclosion, and ACh release from KCs was imaged with GRAB-ACh (*Jing et al., 2020*; *Jing et al., 2018*; *Zhang et al., 2019*). Control flies (containing R13F02-Gal4, UAS-GRAB-ACh, and tub-Gal80$^{ts}$, but lacking a UAS-RNAi) exhibited plasticity across the γ lobe in similar spatial patterns as observed in wild-type animals (↑CS+:CS- in γ1–3 and ↓CS+:CS- in γ5): there was a robust increase in the CS+:CS- and CS+ potentiation in γ1, with a strong trend toward an increase in CS+:CS- in γ2 (p = 0.052), an increase in the CS+ response in γ3, and a trend toward a CS+ decrease in γ5 (p = 0.120)

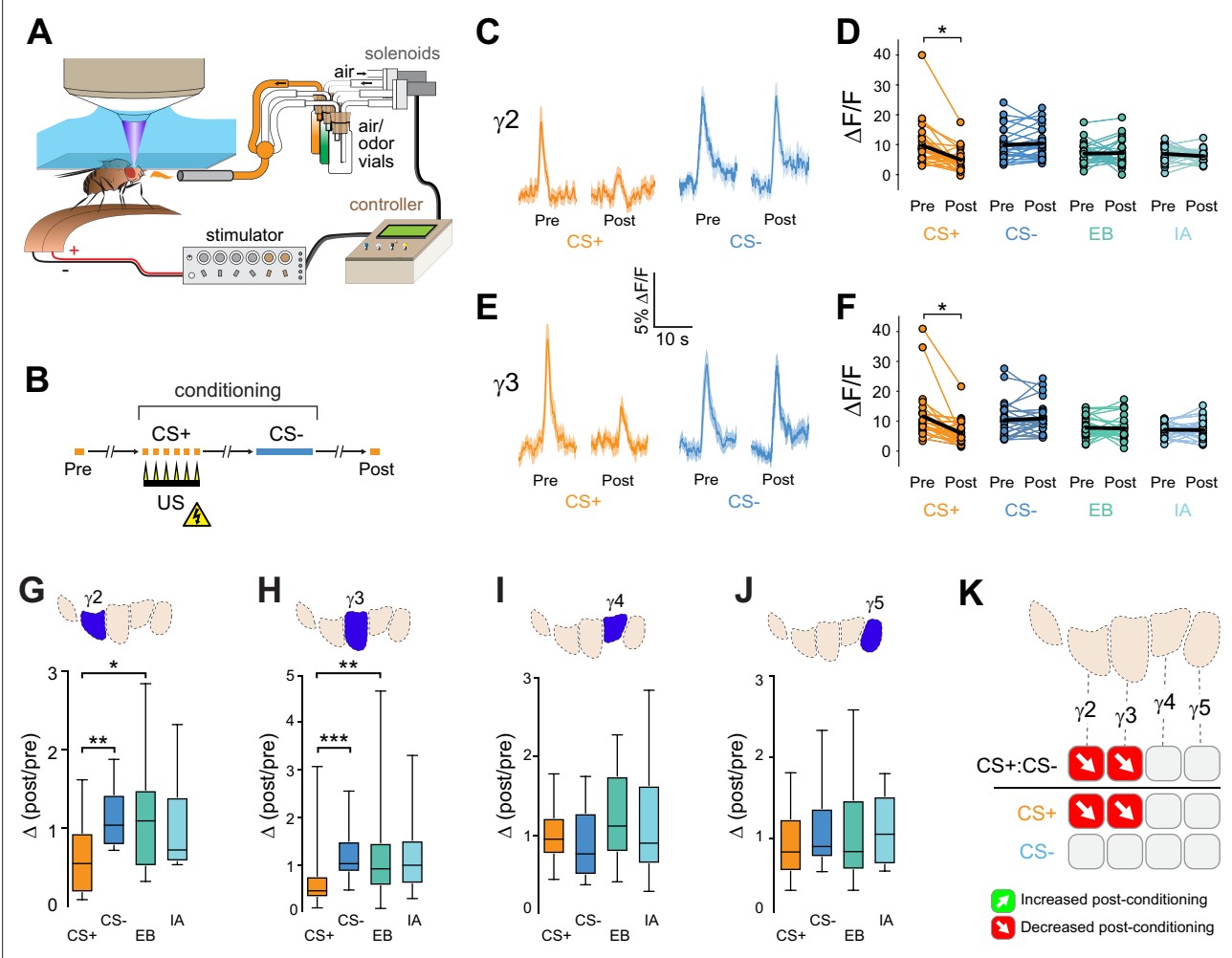

**Figure 2.** Compartment-specific alterations of acetylcholine (ACh) release in the mushroom body (MB) following aversive conditioning. (**A**) Diagram of the aversive conditioning apparatus. (**B**) Aversive conditioning experimental protocol, pairing an odor (the CS+) with an electric shock unconditioned stimulus (US) (six shocks, 60 V). A second odor, the CS-, was presented 5 min after pairing the CS+ and US. One odor was imaged before (Pre) and after (Post) conditioning per animal (CS+ diagrammed here). (**C**) Time series traces showing odor-evoked GRAB-ACh responses pre- and post-conditioning in the γ2 compartment. Responses were imaged to both the CS+ (ethyl butyrate: EB) and CS- (isoamyl acetate: IA) odor, and the line and shading represent the mean ± SEM. (**D**) Quantification of the peak pre- and post-conditioning responses to the CS+ (EB) and CS- (IA) from the γ2 compartment of individual animals, with the mean graphed as a black line. *p<0.05; n = 27 (Wilcoxon rank-sum test). (**E**) Time series traces imaged from the γ3 compartment, graphed as in panel C. (**F**) Quantification of peak responses from the γ3 compartment, graphed as in panel D. (**G–J**) Change in odor-evoked responses (Δ(post/pre) responses), following conditioning (CS+ and CS-) or odor-only presentation (EB and IA). *p<0.05, **p<0.01, ***p<0.001; n = 27 (Kruskal–Wallis/Dunn). (**G**) γ2 compartment. (**H**) γ3 compartment. (**I**) γ4 compartment. (**J**) γ5 compartment. (**K**) Summary of plasticity in ACh release across γ lobe compartments. Red down arrows indicate decreases in the CS+:CS- (first row) or depression of the CS+ relative to odor-only controls (second row).

The online version of this article includes the following figure supplement(s) for figure 2:

**Figure supplement 1.** Effects of aversive conditioning on GRAB-ACh responses across the γ lobe compartments.

(*Figure 3C, E, F*, *Figure 3—figure supplement 1*). When Cac was knocked down conditionally, odor-evoked ACh release was still observed, demonstrating that synaptic exocytosis remained intact. Yet the CS+ potentiation was lost across the γ1–γ3 compartments (*Figure 3D, E, G*, *Figure 3—figure supplement 1*). This demonstrates that the presynaptic Ca$_V$2.1 channel is necessary for potentiation of ACh release induced by learning. Further, this Ca$_V$-dependent potentiation may underlie behavioral learning, a possibility we explore below.

Data from the appetitive conditioning experiments suggested that potentiation of the CS+ response was dependent on the voltage-sensitive Ca$_V$2 Ca$^{2+}$ channel Cac. Interestingly, the trend

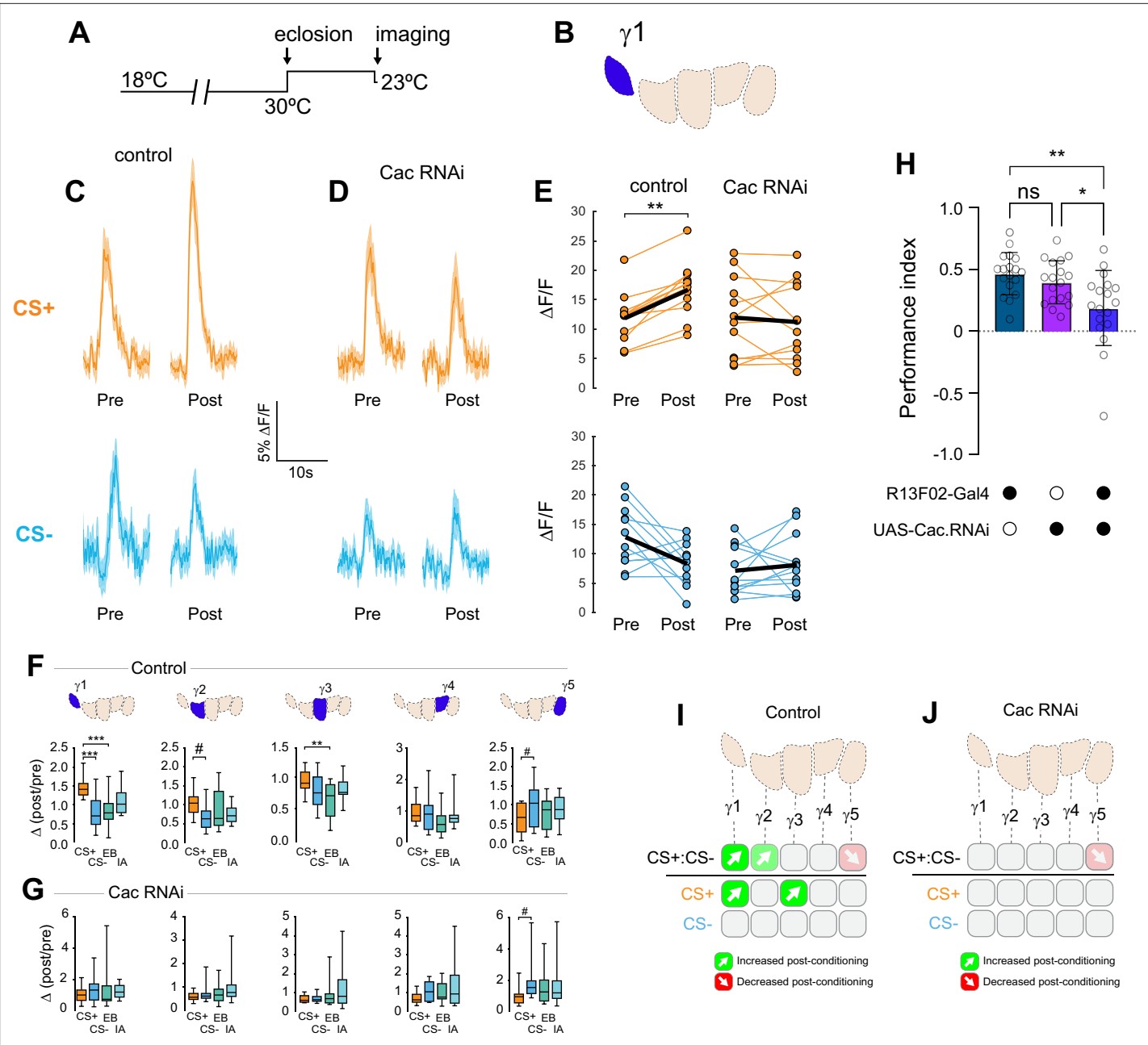

**Figure 3.** Conditional knockdown of the Ca$_V$2 channel Cac in KCs impairs potentiation of ACh release from the MB following appetitive conditioning. (**A**) Diagram of the temperature shifts employed for conditional knockdown of Cac with tub-Gal80ts. (**B**) Diagram of the MB compartments, highlighting the γ1 compartment that was imaged for the data shown in panels C-E. (**C**) Pre- and post-conditioning CS+ (orange; top) and CS- (blue; bottom) odor-evoked ACh release from the γ1 compartment before and after appetitive conditioning, imaged in control animals (w;UAS-GRAB-ACh/+; R13F02-Gal4/UAS-tub-Gal80ts). Time series trace with line and shading representing mean ± SEM. (**D**) Effect of conditional Cac knockdown on odor-evoked ACh responses in the γ1 compartment following appetitive conditioning, graphed as in panel C (genotype: w;UAS-GRAB-ACh/UAS-Cac-RNAi; R13F02-Gal4/UAS-tub-Gal80ts). (**E**) Pre- and post-conditioning ΔF/F CS+ and CS- responses in control and Cac knockdown animals. **p<0.01; n = 12 (Wilcoxon rank-sum test). (**F**) Change in odor-evoked ACh release (Δ(post/pre)) following appetitive conditioning for the CS+, CS-, and odor-only controls (EB and IA) in control animals across the five MB γ lobe compartments: γ1-γ5 (left to right). **p<0.01, ***p<0.001, γ2 compartment #p=0.052, γ5 compartment #p=0.120; n = 12 (Kruskal–Wallis/Dunn). (**G**) Effect of conditional knockdown of Cac on odor-evoked ACh responses across compartments, graphed as in panel F. γ5 compartment #p=0.077; n = 12 (Kruskal–Wallis/Dunn). (**H**) Behavioral appetitive conditioning following conditioning knockdown of Cac. *p<0.05, **p<0.01; n = 12 (ANOVA/Sidak). (**I**) Summary of plasticity in ACh release across γ lobe compartments in control (uninduced) animals. (**J**) Summary of plasticity in ACh release (as in I) in animals with conditional Cac knockdown.

The online version of this article includes the following figure supplement(s) for figure 3:

**Figure supplement 1.** Behavioral avoidance of (**A**) ethyl butyrate (EB) and (**B**) isoamyl acetate (IA).

toward CS+ depression in the most distal γ5 compartment remained intact when Cac was knocked down (**Figure 3G and J**). This suggests that while presynaptic potentiation requires Cac across the MB compartments, depression may not. To directly test whether depression of the CS+ was affected, we turned to aversive conditioning, which generates robust CS+ depression in the proximal γ compartments (**Figure 2**). Cac was knocked down using the same conditional RNAi strategy as above. Control flies for conditional Cac knockdown experiments exhibited similar CS+ depression in the proximal γ2 and γ3 compartments following aversive conditioning (**Figure 4A-C**, **Figure 4—figure supplement 1**). Knockdown of Cac did not appreciably impair depression of CS+ responses (**Figure 4D-F**, **Figure 4— figure supplement 1**). There was a significant depression in γ2, both in terms of the drop in CS+ following conditioning (**Figure 4D**, **Figure 4—figure supplement 1**) and when comparing the Δ(post/ pre) of the CS+ to either the CS- or the odor-only control (EB) (**Figure 4E**, **Figure 4—figure supplement 1**). In the γ1 compartment, there was a trend toward a decrease in the CS+ response compared to the CS- that matched the controls (**Figure 4—figure supplement 1**). In γ3, the difference between the CS+ and CS- (or CS+ vs odor-only control) did not reach significance; yet there was a trend in the same direction as the controls and the CS+ response significantly depressed following conditioning (**Figure 4—figure supplement 1**). Overall, these data suggest that moderate knockdown of Cac does not affect learning-induced depression of ACh release (in contrast to potentiation).

## Reduction of Cac in KCs impairs behavioral reward learning

Reducing Cac levels in KCs altered the pattern of plasticity in ACh release across MB compartments following conditioning, impairing potentiation following appetitive conditioning. This represents a potential physiological mechanism underlying reward learning. To test whether reduction in Cac levels impaired reward learning, we carried out behavioral olfactory appetitive conditioning experiments. Cac was knocked down in KCs using the same conditional expression strategy as above, reducing Cac level ~29% (**Figure 3—figure supplement 1**). Flies were trained with a conditioning paradigm in which a CS+ (EB or IA) was paired with sucrose, the other odor was presented as the CS-, and the flies were allowed to choose between the two odors in a T-maze. Conditional Cac knockdown impaired behavioral performance (**Figure 3H**), reminiscent of its previously reported role in aversive conditioning (**Hidalgo et al., 2021**). Odor avoidance was unaffected by Cac knockdown (**Figure 3—figure supplement 1A**,B), demonstrating that the effect was not due to loss of olfactory acuity. Overall, these findings suggest that the learning-induced facilitation of neurotransmitter release in the proximal γ compartments contributes to behavioral action selection in reward learning.

## Post-conditioning odor contrast and maintenance of odor responses are dependent on IP₃ Signaling

$Ca^{2+}$ release from the endoplasmic reticulum (ER) is a major source of stimulus-evoked $Ca^{2+}$ in neurons, including KCs, and modulates various forms of synaptic/homeostatic plasticity (**Handler et al., 2019**; **James et al., 2019**; **Taufiq et al., 2005**). Therefore, we reasoned that inositol triphosphate receptor (IP₃R) mediated $Ca^{2+}$ release may contribute to presynaptic plasticity across MB compartments. To test this, we conditionally knocked down the IP₃R in the adult MB with RNAi. GRAB-ACh was expressed in the MB (as above), while conditionally knocking down IP₃R (**Figure 4**, **Figure 4—figure supplement 1**, **Figure 4—figure supplement 2**). For these experiments, flies were aversively conditioned (IP₃R knockdown impairs feeding under the microscope, precluding appetitive conditioning). Knockdown of IP₃R eliminated the post-conditioning contrast between the CS+ and CS- (i.e. the difference between the CS+ and CS-) (**Figure 4G-I**, **Figure 4—figure supplement 1**). However, this was not due to a loss of depression in the CS+ groups, rather it was due to the appearance of olfactory adaptation in the other groups. This was seen as a decrease in the post-treatment odor responses in the CS- and odor-only controls (**Figure 4G**, **Figure 4—figure supplement 1C**), with concomitant Δ(post/ pre) values < 1 (**Figure 4H**, **Figure 4—figure supplement 1D**). As there was no US presented in the odor-only control groups, this reduction in olfactory response represents adaptation, rather than a conditioned, associative change. Therefore, in normal conditions, release of $Ca^{2+}$ from the ER via IP₃R is necessary to maintain odor responsivity upon repeated odor presentations. Loss of IP₃R renders the KCs more susceptible to adaptation at this time point (**Figure 1—figure supplement 1**, **Figure 4H**), reducing the contrast between the CS+ (which exhibits depression following aversive learning) and the other odor(s).

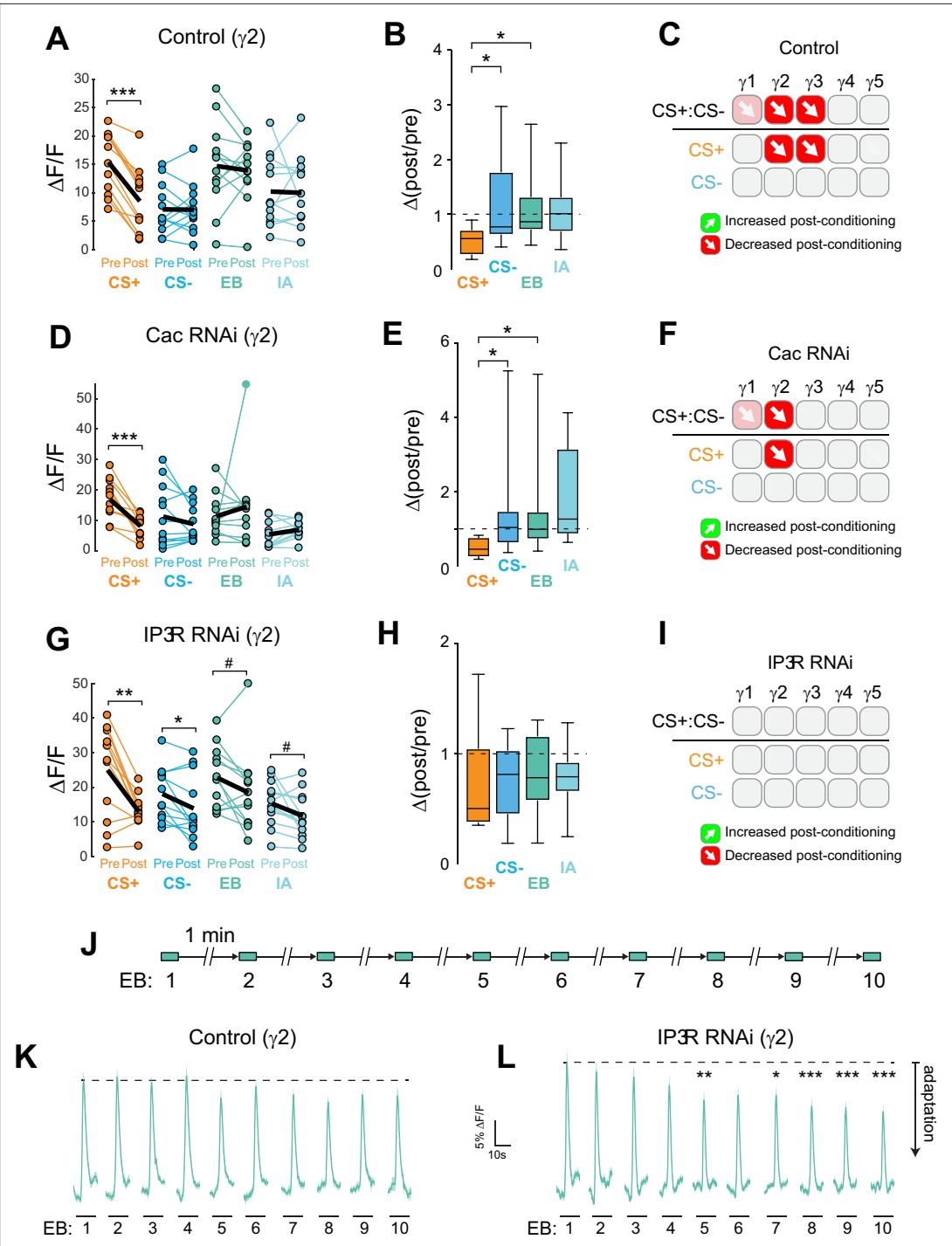

**Figure 4.** Cac and IP$_3$R exert distinct effects on synaptic plasticity and maintenance of olfactory responses in the γ2 compartment following aversive conditioning. (**A,D,G**) Quantification of odor-evoked responses in control, Cac RNAi, and IP$_3$R RNAi animals, respectively. *p<0.05, **p<0.01, ***p<0.001; n = 12 (Wilcoxon rank-sum test). IP$_3$R RNAi: EB #p=0.092, IA #p=0.064. (**B,E,H**) Change in odor-evoked ACh release (Δ(post/pre) responses) following conditioning for the CS+, CS-, and odor-only controls (EB and IA). *p<0.05; n = 12 (Kruskal–Wallis/Dunn). (**C,F,I**) Summary of plasticity in ACh release across γ lobe compartments in control, Cac RNAi, and IP$_3$R RNAi animals, respectively. Red down arrows indicate decreases in the CS+:CS- (first row) or depression of the CS + relative to odor-only controls (second row). (**J**) Olfactory adaptation protocol. (**K**) Odor-evoked ACh release (ΔF/F time series traces) measured across repeated odor presentations in control flies. (**L**) Odor-evoked ACh release (ΔF/F time series traces) measured across repeated odor presentations in IP$_3$R knockdown flies. *p<0.05, **p<0.01, ***p<0.001; n = 27 (two-way ANOVA/Sidak).

The online version of this article includes the following figure supplement(s) for figure 4:

*Figure 4 continued on next page*

*Figure 4 continued*

**Figure supplement 1.** Effects of aversive conditioning on GRAB-ACh responses across the γ lobe compartments using GRAB-ACh with control, Cac RNAi and IP$_3$R RNAi flies.

**Figure supplement 2.** Adaptation of odor-evoked GRAB-ACh responses across the γ lobe compartments in control and IP$_3$R knockdown flies upon repeated odor presentation.

**Figure supplement 3.** Effect of silencing KCs on GRAB-ACh responses across the γ lobe compartments.

## IP$_3$R maintains neurotransmitter release during repeated odor exposure

To directly test how adaptation to odors was impacted by the loss of IP$_3$R, as suggested by analysis of pre/post odor responses in the conditioning experiments above, we carried out an olfactory adaptation assay. Flies were presented an odor 10 times over the course of 10 min while imaging the GRAB-ACh responses. To test for olfactory adaptation at each time point, the ΔF/F was compared to the initial naïve odor presentation (*Figure 4J*, *Figure 4—figure supplement 1*). Control animals exhibited no significant olfactory adaptation across the 10 trials in any compartment (*Figure 4K*, *Figure 4—figure supplement 1*). In contrast, knocking down IP$_3$R in KCs produced significant olfactory adaptation in the γ2 and γ3 compartments (*Figure 4L*, *Figure 4—figure supplement 1*). In γ2, following the fourth-odor presentation, there was a significant depression in odor-evoked ACh release at all-time points save one (*Figure 4L*, *Figure 4—figure supplement 1*). A similar effect was observed in the γ3, where following the third-odor presentation, a significant depression in release occurred (*Figure 4—figure supplement 2*). The γ1 compartment showed no evidence of adaptation, suggesting that IP$_3$R may support plasticity in this compartment via another mechanism (*Figure 4—figure supplement 2*). The γ4 and γ5 compartments exhibited no significant adaptation (*Figure 4—figure supplement 2*). Overall, these data suggest that IP$_3$R-dependent Ca$^{2+}$ release from the ER may contribute to maintenance of synaptic strength, preventing adaptation during repeated stimuli. Further, loss of IP$_3$R leads to alterations in neurotransmitter release that particularly impact compartments dominated by CS+depression.

To confirm the site of ACh release, we blocked neurotransmitter release from KCs by expressing the temperature-sensitive dynamin mutant *Shibire*[ts] with R13F02-Gal4 (*Kitamoto, 2001*; *McGuire et al., 2001*). Odor-evoked GRAB-ACh transients were imaged across MB compartments, as above, before, and after shifting flies from the permissive (20°C) to the restrictive (30°C) temperature. Blocking synaptic transmission from KCs inhibited the GRAB-ACh responses across compartments (*Figure 4—figure supplement 3*). There was some residual response at the restrictive temperature (particularly in γ1), which we reasoned may emanate from KCs that were not labeled by the R13F02-Gal4 driver. To quantify the coverage of this driver, we counted GRAB-ACh labeled neurons (immunostained with an anti-GFP antibody), finding that R13F02-Gal4 labeled 745 ± 47 neurons (*Figure 4—figure supplement 3*) out of ~2000 total KCs (*Aso et al., 2009*). Inhibiting this relatively small subset of KCs significantly reduced the GRAB-ACh response from the MB compartments, demonstrating that KCs are the major ACh source, though cholinergic MBONs could also contribute to the signal (*Scheffer et al., 2020*).

## Compartmentalized plasticity propagates into downstream mushroom body output neurons

Since ACh release from each compartment provides input to unique postsynaptic MBONs, the presynaptic plasticity observed in each compartment should be mirrored in the respective postsynaptic MBON(s) innervating that compartment. To test this, we imaged Ca$^{2+}$ responses in MBONs with GCaMP and examined the effect of appetitive conditioning. Four sets of MBONs were tested, each innervating and receiving cholinergic input from a distinct MB γ lobe compartment: γ1pedc>α/β, γ2α′1, γ3/γ3β′1, and γ5β′2a (*Figure 5A*). Within the γ lobe, these neurons innervate the γ1, γ2, γ3, and γ5 compartments, respectively (*Figure 5B–F*). We focused our analysis on the γ2α′1 and γ3/γ3β′1 MBONs, which have not been studied intensively in the context of appetitive conditioning. The γ2α′1 MBON exhibited an increase in the relative CS+ responses following conditioning (*Figure 5D*). The plasticity could not be unambiguously attributed to purely CS+ potentiation or CS- depression. The γ3/γ3β′1 MBONs exhibited an increase in the relative CS+ responses (*Figure 5F*). In this case, it was due to potentiation of the CS+ response (*Figure 5F*). Note that this pair of MBONs is not parsed with

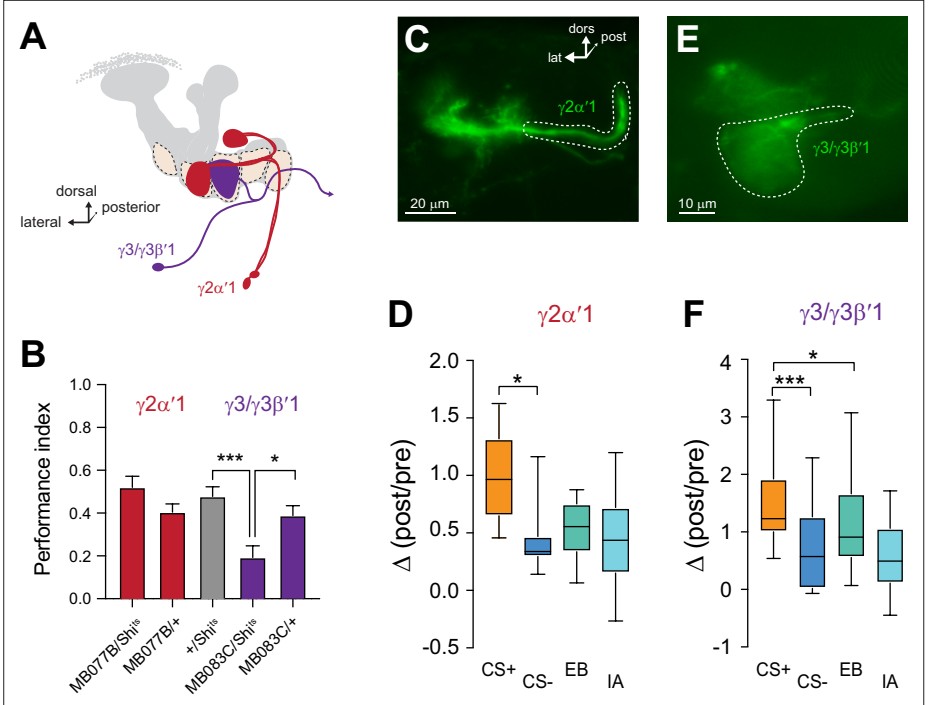

**Figure 5.** Plasticity in MBON Ca²⁺ responses mirrors compartmental plasticity in the mushroom body (MB) neurons. (**A**) Diagram of MBONs innervating specific γ lobe compartments, viewed from a frontal plane. Each MBON is bilaterally paired, though only one is drawn here for visual clarity. (**B**) Behavioral appetitive conditioning when synaptic output from the γ2α′1 and γ3 MBONs was blocked with Shibire^ts (Shi^ts), driven with the MB077B and MB083C split Gal4 drivers, respectively. *$p<0.05$; ***$p<0.001$; n = 16 (ANOVA/Sidak). (**C,E**) Representative confocal images of the γ2α′1 and γ3 MBONs, respectively. The region of interest circumscribed for quantification is drawn with a dotted white line. lat: lateral, dors: dorsal, post: posterior. (**D,F**) Change in odor-evoked responses (Δ(post/pre responses)) in the γ2α′one and γ3 MBONs, respectively, following appetitive conditioning. *$p<0.05$; ***$p<0.001$; n = 12 (Kruskal–Wallis/Dunn).

The online version of this article includes the following figure supplement(s) for figure 5:

**Figure supplement 1.** Plasticity in MBON Ca²⁺ responses mirrors compartmental plasticity in the MB neurons.

available drivers and they were imaged together. One of them (γ3β′1) receives major input from the β′1 lobe/compartment in addition to γ3. Presynaptically, the γ3 compartment exhibited a depression in the CS- response, suggesting that the potentiation in the γ3/γ3β′1 CS+ response may emanate from potentiated β′1 inputs. The γ1pedc>α/β and γ5β′2 a MBONs have been reported to exhibit bidirectional plasticity following aversive and appetitive conditioning (*Owald et al., 2015*; *Perisse et al., 2016*); we examined the plasticity in these MBONs (and others) using the same odors/conditions that were used to probe presynaptic plasticity in KCs. Following appetitive conditioning, the γ1pedc>α/β MBON exhibited a significant elevation of the CS+ response relative to the CS- (*Figure 5—figure supplement 1*). This was due to a potentiation of the CS+ response, as the post-conditioning CS+ response was significantly larger than the corresponding odor-only control. Finally, appetitive conditioning produced plasticity in the opposite direction in the γ5β′2 a MBON; this neuron exhibited a decrease in the CS+ response relative to the CS- (*Figure 5—figure supplement 1*). Overall, the directionality of the plasticity in MBONs matched that observed in ACh responses in the presynaptic compartment. Thus, compartmentalized, presynaptic plasticity in neurotransmitter release from the MB compartments likely plays a role in modulating the MBON responses following learning.

## Synaptic activity from the γ3-innervating MBONs mediates reward learning

The unique role of the γ2 and γ3 compartments in encoding CS- plasticity, as well as their strong plasticity following both appetitive and aversive conditioning, led us to question the behavioral roles of

the MBONs that receive input from these compartments (*Figures 1 and 5A*). In particular, the involvement of these MBONs in appetitive memory is unclear. To test whether the MBONs innervating the γ2 and γ3 compartments mediate short-term appetitive memory, we carried out behavioral appetitive classical conditioning, blocking synaptic transmission from MBONs with *Shibire*ts (*Shi*ts) (*Kitamoto, 2001*; *McGuire et al., 2001*; *Figure 5A and B*). Blocking the γ2α′1 MBON did not significantly impair performance in appetitive conditioning (*Figure 5B*). Therefore, while activation of the γ2α′1 MBON drives approach behavior (*Aso et al., 2014b*) and the neuron is necessary for aversive memory (*Berry et al., 2018*), it is not crucial for appetitive short-term memory in the otherwise intact nervous system. In contrast, blocking synaptic transmission from the γ3/γ3β′1 MBONs significantly impaired appetitive conditioning performance (*Figure 5B*). This demonstrates that the output of the γ3/γ3β′1 MBONs is not only sufficient for protein synthesis-dependent appetitive long-term memory (*Wu et al., 2017*) but also serves a key role for normal appetitive short-term memory. These neurons convey the output of the MB γ3 compartment to the crepine and superior medial protocerebrum (where they innervate interneurons that project to the fan-shaped body and lateral accessory lobe further downstream), as well as provide direct contralateral MB feedback and form polysynaptic feedback loops via MB-innervating PAM dopaminergic neurons and other MBONs (*Scaplen et al., 2021*; *Xu et al., 2020*). The γ3 PAM dopaminergic neuron is also bidirectionally modulated through sucrose-activated allatostatin A neurons (*Yamagata et al., 2016*). These multi-layered connections provide several routes through which they could modulate behavioral output following learning. Overall, the present data suggest that the γ3/γ3β′1 MBONs receive input from an MB compartment with unique physiology and represent a key node through which discriminative effects influence sucrose-activated appetitive memory and decision-making.

## Discussion

Compartmentalized plasticity in neurotransmitter release expands the potential computational capacity of learning circuits. It allows a set of odor-coding MB neurons to bifurcate their output to different downstream approach- and avoidance-driving downstream output neurons, independently modulating the synaptic connections to alter action selection based on the conditioned value of olfactory stimuli. The KCs modify the encoded value of olfactory stimuli through bidirectional plasticity in odor responses, which vary in a compartment-specific manner along the length of the axons. These changes were observed following pairing an olfactory CS with gustatory/somatosensory US (sucrose feeding or electric shock) in vivo. The CS+ and CS- drive unique patterns of plasticity in each compartment, demonstrating that olfactory stimuli are reweighted differently across compartments following learning, depending on the temporal associations of the stimuli. Different molecular mechanisms govern the potentiation of trained odor responses ($Ca_V2$/Cac) and maintenance of responsivity over time ($IP_3R$). Finally, one set of γ output neurons, γ3/γ3β′1, is important for appetitive short-term memory.

Learning-induced plasticity of ACh release in the MB was bidirectional within the compartment, depending on the valence of the US, and was coherent with the valence of the MBON downstream of the compartment. Notably, the γ2 and γ3 MB compartments, which relay information to approach-promoting MBONs (*Aso et al., 2014b*), exhibited plasticity that was coherent with promoting behavioral approach following appetitive conditioning and avoidance after aversive conditioning. There was an increase in the relative CS+:CS- ACh responses after appetitive conditioning, and conversely reduced CS+:CS- ACh responses following aversive conditioning. For this study, we focused on the time point 5 min following conditioning, which is consistent with behavioral short-term memory. Aversive conditioning was previously reported to decrease neurotransmitter release from KCs (*Zhang and Roman, 2013*; *Zhang et al., 2019*). Indirect evidence, via $Ca^{2+}$ imaging in presynaptic KCs, suggested that increases in presynaptic neurotransmission could also be associated with learning. Pairing odor with stimulation of appetitive PAM dopaminergic neurons potentiates odor-evoked cytosolic $Ca^{2+}$ transients across the KC compartments (*Boto et al., 2014*). Appetitive conditioning increases odor-evoked $Ca^{2+}$ transients across KC compartments (*Louis et al., 2018*). Stimulation of dopaminergic circuits associated with reward learning potentiate MB γ4 connections with the respective γ4 MBON (*Handler et al., 2019*). We did not detect a statistically significant effect in γ4 with appetitive or aversive classical conditioning, though the CS+ and CS- trended in the same direction as the adjacent γ5 compartment following conditioning. Overall, the present data demonstrate that there are

bidirectional changes in neurotransmitter (ACh) release from MB compartments following appetitive vs aversive learning and provide a window into the spatial patterns of plasticity across compartments following associative learning.

Behavioral alterations following conditioning involve changes in responses among the MBONs. As the KCs provide presynaptic olfactory input to the MBONs, it was a logical a priori assumption that presynaptic plasticity in the KCs could be altered in a compartmental manner and contribute to the changes in MBON responses after conditioning. Yet data from previous $Ca^{2+}$ imaging experiments have not completely supported this model. Compartmentalized effects have been observed in KCs with non-associative learning protocols (*Cohn et al., 2015*) and within the γ4 compartment following associative learning (*Handler et al., 2019*). In contrast, classical conditioning produces no compartmentalized differences in odor-evoked $Ca^{2+}$ responses. Appetitive conditioning with odor + sucrose pairing increases odor-evoked cytosolic $Ca^{2+}$ transients in KCs across the γ lobe compartments (*Louis et al., 2018*). Aversive conditioning produces no net change across the compartments (*Louis et al., 2018*), but alters synapse-specific $Ca^{2+}$ responses at the individual bouton level (*Bilz et al., 2020*). If the compartmental effects of conditioning (observed with $Ca^{2+}$ imaging) in KCs drove a proportional change in neurotransmitter release, both the approach- and avoidance-promoting MBONs would be simultaneously potentiated. Extracellular influx of $Ca^{2+}$ through voltage-gated calcium channels is a primary driver of neurotransmitter release; however, there are multiple sources of $Ca^{2+}$ in the cytosol that could contribute to the GCaMP signals (*Grienberger and Konnerth, 2012*). A major conclusion of the present study is that learning drives compartmentalized plasticity in neurotransmitter release that is coherent with the behavioral valence of the corresponding MBON.

At least two major molecular mechanisms govern the spatial patterns of plasticity across the MB compartments: a Cac-dependent CS+ potentiation and an $IP_3R$-dependent maintenance of sensory responses over trials/time. This suggests that different sources of $Ca^{2+}$ play different roles in regulating KC synaptic responses. Cac is the pore-forming subunit of the voltage-sensitive, presynaptic $Ca_V2$ $Ca^{2+}$ channel in *Drosophila*. $Ca_V2$ channels regulate several forms of synaptic plasticity, including paired-pulse facilitation, homeostatic plasticity, and long-term potentiation (*Frank et al., 2006*; *Inchauspe et al., 2004*; *Nanou et al., 2016*). Our data suggest that these channels regulate the spatial patterns of learning-induced plasticity in the MB unidirectionally (from baseline), with Cac underlying potentiation but not depression. $Ca_V2$ channel activity is modulated by presynaptic calcium and G protein-coupled receptor activity (*Zamponi and Currie, 2013*), and channel localization in the active zone dynamically regulates synaptic strength (*Gratz et al., 2019*; *Lübbert et al., 2019*). Thus, Cac insertion into, or increased clustering within, the active zones may underlie learning-induced potentiation (e.g. in the γ1-γ2 compartments following appetitive conditioning). Conditional knockdown of Cac, which reduced Cac levels by ~29%, impaired this potentiation, likely by decreasing the number of available channels for modulation. Baseline stimulus-evoked neurotransmitter release was maintained during Cac knockdown, mediated either by the significant residual Cac expression or compensation by other intracellular $Ca^{2+}$ channels/sources. In contrast to the Cac effect on potentiation, $IP_3R$ was necessary to maintain normal odor responsivity when odors were presented repeatedly across multiple trials (whether those were pre/post trials in the conditioning protocol or 10× odor presentations in the adaptation protocol). This is broadly consistent with the temporal role of $IP_3R$ in maintenance of presynaptic homeostatic potentiation at the neuromuscular junction (*James et al., 2019*). In addition, dopaminergic circuits associated with reward learning drive release of $Ca^{2+}$ from the endoplasmic reticulum when activated with KCs in a backward pairing paradigm ex vivo, potentiating MB γ4 connections with the respective γ4 MBON (*Handler et al., 2019*). This is consistent with a role for ER calcium in positively regulating synaptic strength.

We observed potentiation and depression of ACh release across multiple MB compartments following conditioning, providing a presynaptic mechanism that potentially contributes to shaping conditioned MBON responses. Importantly, by comparing the CS+ and CS- responses to those of untrained odors, we ascribed plasticity to potentiation or depression (accounting for any non-associative olfactory adaptation) within each compartment. This is relevant for modeling efforts, where it has been unclear whether to include potentiation (along with depression) in the learning rule(s) at KC-MBON synapses (*Abdelrahman et al., 2021*; *Bennett et al., 2021*; *Jiang and Litwin-Kumar, 2021*; *Springer and Nawrot, 2021*). In addition, the experiments revealed an additional layer of spatial regulation in the γ1–γ3 compartments: a gradient of CS+ potentiation to CS- depression

following appetitive conditioning. Specifically, the CS+/CS- relationship changed in a linear gradient down the γ1–γ3 compartments following appetitive conditioning. Appetitive conditioning increased CS+ responses in the γ1 compartment, while decreasing the CS- responses in the γ3 compartment. The γ2 compartment yielded a mix of these responses. These patterns of plasticity have the net effect of increasing the relative response to the CS+ odor (↑CS+:CS-). Since the MBONs postsynaptic to these compartments drive behavioral approach (*Aso et al., 2014b*), this would bias the animal to approach the CS+ if it encountered both odors simultaneously. Such a situation occurs at the choice point of a T-maze during retrieval in a classical conditioning assay. The CS+ and CS- produce different patterns of plasticity at different loci (e.g. γ1 vs γ3), which presumably coordinate to regulate behavior via temporal integration of the odor and US cues (*Berry et al., 2018*; *Handler et al., 2019*; *Jacob and Waddell, 2020*; *König et al., 2018*; *Tanimoto et al., 2004*; *Tully and Quinn, 1985*). The CS+ is temporally contiguous with the US, while the CS- is nonoverlapping. Therefore, the timing of CS/US pairing drives plasticity differently in each compartment. These patterns of plasticity presumably coordinate to regulate memory formation and action selection during retrieval. For instance, while the γ1–γ3 compartments exhibited ↑CS+:CS- following appetitive conditioning, the γ5 compartment exhibited plasticity in the opposite direction: decreasing the relative response to the CS+ odor (↓CS+:CS-). As the γ5 compartment is presynaptic to an avoidance-promoting MBON, this plasticity pattern would coherently contribute to biasing the animal toward CS+ approach (reducing CS+ avoidance). Thus, it would work in concert with the plasticity in γ1–γ3 to bias the animal toward behavioral approach. Overall, plasticity is regulated in each MB compartment individually by the timing of events and the valence of the US, with the changes coordinated across multiple compartments to coherently drive behavior.

Behaviorally, MBONs innervating the γ lobe variably drive behavioral approach or avoidance when stimulated (*Aso et al., 2014b*). Despite the approach-promoting valence of the γ2α'one and γ3/γ3β'1 MBONs, among them, only the γ3/γ3β'1 MBONs produced a loss-of-function phenotype in behavioral appetitive conditioning. This suggests that redundancy and/or different weighting across approach promoting MBONs renders the system resilient to silencing some of them. A previous study found effects of blocking the γ2α'1 MBONs, though not γ3/γ3β'1 MBONs, when blocking individual steps of memory processing (acquisition, retention, and/or retrieval) with a 1 hr appetitive memory protocol (*Ichinose et al., 2021*). This suggests that the different MBONs have differing roles across time, with some redundancy in appetitive processing. Blocking synaptic output of γ3/γ3β'1 MBONs reduced appetitive conditioning performance in our experiments immediately following conditioning, suggesting that these neurons play a specific role in appetitive short-term memory.

The present and previous studies suggest that alterations of MBON activity following learning are the product of both presynaptic and postsynaptic plasticity at the KC-MBON synapses, as well as feedforward inhibition (*Hige et al., 2015a*; *Perisse et al., 2016*; *Pribbenow et al., 2021*). Blocking synaptic output from KCs impairs the acquisition of appetitive memories (30–60 min after conditioning), suggesting a role for postsynaptic plasticity (*Pribbenow et al., 2021*; *Schwaerzel et al., 2003*). However, this does not rule out presynaptic plasticity, as blocking KC output (with R13F02) leaves signaling from reinforcing dopaminergic neurons partially intact (*Cervantes-Sandoval et al., 2017*), which likely shapes the presynaptic KC responses via heterosynaptic plasticity. At the circuit level, polysynaptic inhibition can convert depression from select MB compartments into potentiation in MBONs following learning; in one established example, reduction of odor-evoked responses in the GABAergic γ1pedc MBON following aversive conditioning disinhibits the downstream γ5β'2 a MBON (*Owald et al., 2015*; *Perisse et al., 2016*).

KC-MBON synapses represent one node of learning-related plasticity, which is distributed across multiple sites during learning. Short-term memory-related plasticity has been observed in multiple olfactory neurons, such as the antennal lobes (*Yu et al., 2004*) and KCs (*Louis et al., 2018*; *Wang et al., 2008*). In addition, connectomics studies have revealed complex connectivity within and beyond the MB, which is a multi-layered network including circuit motifs that influence the propagation of information and generation of plasticity during learning. Such connections include recurrent feedback (*Aso et al., 2014a*; *Ichinose et al., 2015*; *Li et al., 2020*; *Otto et al., 2020*; *Scaplen et al., 2021*; *Takemura et al., 2017*; *Zhao et al., 2018*). Some of these recurrent connections are from cholinergic MBONs that synapse within the MB, which could have contributed to the ACh signals we observed in this study. For instance, the γ2α'1 MBON is a cholinergic MBON that sends ~6% of its

output back to the γ lobe (*Scheffer et al., 2020*). Some of the recurrent connections are formed by dopaminergic neurons, such as the PAM γ4<γ1/γ2 (*Aso et al., 2014a*; *Li et al., 2020*). In addition, reciprocal connections between KCs and dopaminergic neurons in the vertical lobes are necessary for memory retrieval (*Cervantes-Sandoval et al., 2017*). This adds another layer of recurrent circuitry that may participate in reinforcement during associative learning (*Aso et al., 2014b*; *Ichinose et al., 2015*; *Ichinose et al., 2021*). Across these circuits, some neurons corelease several neurotransmitters and act on an array of postsynaptic receptors, which contribute to plasticity distributed across multiple sites (*Aso et al., 2019*; *Barnstedt et al., 2016*; *Berry et al., 2012*; *Bielopolski et al., 2019*; *Cohn et al., 2015*; *Handler et al., 2019*; *Haynes et al., 2015*; *Keene et al., 2004*; *Kim et al., 2007*; *Liu and Davis, 2009*; *Pribbenow et al., 2021*; *Schroll et al., 2006*; *Séjourné et al., 2011*; *Silva et al., 2015*; *Wu et al., 2013*; *Zhou et al., 2019*).

Overall, plasticity between KCs and MBONs may guide behavior through biasing network activation to alter action selection in a probabilistic manner. Appetitive conditioning drives compartmentalized, presynaptic plasticity in KCs that correlates with postsynaptic changes in MBONs that guide learned behaviors. Prior studies documented only depression at these synapses at short time points following conditioning (*Hige et al., 2015a*; *Zhang and Roman, 2013*; *Zhang et al., 2019*). Here, we observed both potentiation and depression in ACh release in the MB, suggesting that bidirectional presynaptic plasticity modulates learned behaviors. These bidirectional changes likely integrate with plasticity at downstream circuit nodes that also undergo learning-induced plasticity to produce network-level alterations in odor responses across the olfactory pathway following salient events. Thus, plasticity in ACh release from KCs functions to modulate responsivity to olfactory stimuli features across graded plasticity maps down the MB axons.

## Materials and methods

### Fly strains

Flies were fed and maintained on a standard cornmeal agar food mixture on a 12:12 light:dark cycle. The 238Y-Gal4 driver was selected for maximal KC coverage (*Aso et al., 2009*) and high expression levels (*Louis et al., 2018*). The R13F02-Gal4 driver was selected for greater specificity in KCs (*Jenett et al., 2012*). MBON drivers were selected from the FlyLight and split-Gal4 collections (R12G04, MB077b, and MB083c) (*Jenett et al., 2012*; *Pfeiffer et al., 2010*). The γ5β′2 a LexA MBON driver was generated by Krystyna Keleman (*Zhao et al., 2018*). RNAi lines were obtained from the VDRC (Cac: 101478) (*Dietzl et al., 2007*) and TRiP collections (IP$_3$R: 25937) (*Perkins et al., 2015*) and crossed into flies expressing R13F02-Gal4 and tub-Gal80$^{ts}$ (*McGuire et al., 2003*). Final experimental genotypes were: Cac (w;UAS-GRAB-ACh/UAS-Cac-RNAi;R13F02-Gal4/UAS-tub-Gal80$^{ts}$) and IP$_3$R (w;UAS-GRAB-ACh/UAS-tub-Gal80$^{ts}$;R13F02-Gal4/UAS-IP3R-RNAi), compared to genetic controls (w;UAS-GRAB-ACh/+;R13F02-Gal4/UAS-tub-Gal80$^{ts}$). For quantitative analysis of Cac knockdown, flies of the experimental genotype (w;UAS-Cac-RNAi/UAS-tub-Gal80$^{ts}$;tub-Gal4/+) were compared to genetic controls (w;UAS-tub-Gal80$^{ts}$/VIE-260B;tub-Gal4/+).

### Fly preparation for in vivo imaging

Flies were briefly anesthetized, placed in a polycarbonate imaging chamber (*Tomchik, 2013*), and fixed with myristic acid (Sigma-Aldrich). The proboscis was fixed in the retracted position, except for appetitive conditioning experiments (as noted below). A cuticle window was opened, and the fat and tracheal air sacs were carefully removed to allow optical access to the brain. The top of the chamber was filled with saline solution (103 mM NaCl, 3 mM MBl, 5 mM HEPES, 1.5 mM CaCl$_2$, 4 mM MgCl2·6H$_2$O, 26 mM NaHCO$_3$, 1 mM NaH$_2$PO$_4$·H2O, 10 mM trehalose, 7 mM sucrose, and 10 mM glucose), which was perfused over the dorsal head/brain at 2 ml/min via a peristaltic pump.

### In vivo imaging

GRAB-ACh (*Jing et al., 2020*; *Jing et al., 2018*; *Zhang et al., 2019*) was driven in the KCs, using the 238Y or R13F02 driver. Within the KCs, ROIs were drawn around five γ lobe compartments (γ1–5) within a single imaging plane for appetitive, and (γ2–5) for aversive. Imaging was performed with a Leica TCS SP8 confocal microscope utilizing appropriate laser lines and emission filter settings. Odors were delivered with an airstream for 1 s (60 ml/min flow rate) by directing the air flow with solenoid

valves between an empty vial (air) to another containing 1 µl odorant spotted on filter paper. Odor-evoked responses were calculated as the baseline normalized change in fluorescence ($\Delta F/F$), using the maximum $\Delta F/F$ within a 4 s window after odor delivery. In experiments with RNAi, flies containing the R13F02-Gal4, GRAB-ACh, a UAS-RNAi line, and tub-Gal80$^{ts}$ were constructed; flies were raised at 18°C until eclosion, then transferred to 32°C for 4–10 days prior to the experiment. Experiments were carried out at room temperature (23°C) for ACh imaging/conditioning. For Ca$^{2+}$ imaging experiments, GCaMP6f was expressed in the MBONs using the R12G04 (γ1pedc), MB077b (γ2α′1), MB083c (γ3) Gal4/split-Gal4 drivers, or the VT014702-LexA (γ5β′2) driver. Regions of interest were selected in accordance with prior studies (*Berry et al., 2018*; *Jacob and Waddell, 2020*; *Zhao et al., 2018*). Experiments were carried out the same way as ACh imaging, except presenting a 3 s odor delivery.

## Appetitive conditioning and imaging

Appetitive conditioning was carried out as previously described (*Louis et al., 2018*). Flies were starved for a period of 18–24 hr prior to conditioning. One odor (the CS+; ethyl butyrate) was presented in conjunction with a paired sucrose (1 M, containing green food coloring) US, and a second odor (the CS-; isoamyl acetate) was presented 30 s later. Six 1 s odor pulses were presented during conditioning over a 30 s period, with a 5 s inter-pulse interval, to prevent desensitization of the reporter. In odor-only control cohorts, the sucrose US was omitted. During conditioning, the US was presented continuously for 30 s. Pre/post odor-evoked responses were imaged prior to and after the imaging protocol, using a 1 s (ACh imaging) or 3 s (Ca$^{2+}$ imaging) odor pulse. During odor-evoked response imaging, proboscis extension was blocked utilizing a thin metal loop attached to a custom motorized micromanipulator. During conditioning, the proboscis was released, and the flies were presented sucrose through a metal pipette fed by a syringe pump controlled via a micro-controller (Arduino). To assess feeding, flies were monitored using a digital microscope (Vividia); sucrose ingestion was visually confirmed by the presence of green food coloring in the abdomen.

## Aversive conditioning and imaging

Flies were mounted in a polycarbonate imaging chamber such that the brain could be imaged while odors were delivered to the antennae and electric shocks delivered to the legs via a shock grid below the fly. Conditioning was carried out by pairing a CS+ odor with electric shocks as follows: 6 × 1 s odor pulses, with a 5 s inter-pulse interval, paired with 6 × 90 V electric shocks. This was followed 30 s later by presentation of 6 × 1 s pulses of the CS- odor with 5 s inter-pulse interval. Pre- and post-conditioning odor-evoked GRAB-ACh responses were imaged using a 1 s odor pulse. In each animal, either the CS+ or CS- odor was tested pre- and post-conditioning.

## Olfactory adaptation protocol

Flies were mounted in the imaging chamber and the proboscis was fixed in place with myristic acid to minimize brain movement. Odors were presented for a 1 s period in 1 min intervals for 10 min. Odor-evoked ACh responses were imaged, the peak $\Delta F/F$ of each response quantified, and the response at each time point was compared to the initial naïve response.

## Behavioral appetitive conditioning

Adult flies, 2–5 day old, were trained under dim red light at 75% relative humidity. Appetitive conditioning experiments were performed in animals starved 16–20 hr. Groups of ~60 flies were exposed for 2 min to an odor (the CS-), followed by 30 s of air and 2 min of another odor, the (the CS+), paired with a 2 M sucrose solution dried on filter paper, at 32°C for *Shibire*$^{ts}$ blockade. The odors were ethyl butyrate and isoamyl acetate, adjusted so that naïve flies equally avoided the two odors (0.05–0.1%). Memory was tested by inserting the trained flies into a T-maze, in which they chose between an arm containing the CS+ odor and an arm containing the CS- odor. Odors were bubbled at 500 ml/min air flow rate. Flies were allowed to distribute for a 2 min choice period. The Performance Index (P.I.) was calculated as (flies in the CS- arm)-(flies in the CS+ arm)/(total flies in both arms). Odor avoidance was tested by putting groups of ~60 flies in a T-maze in which they chose between an arm containing an odor and an arm containing air; an Avoidance Index was calculated as above for the P.I.

## Immunohistochemistry

About 5–7-day old adult flies were dissected in 1% paraformaldehyde in S2 medium and processed as previously described (*Jenett et al., 2012*). Brains were incubated with primary antibodies for 3 hr at

room temperature, followed by secondary antibodies for 3 hr at room temperature and 4 days at 4°C. Incubations were performed in blocking serum (3% normal goat serum). Brains were then stained with NeuroTrace 530/615 Red Fluorescent Nissl Stain (1:100, Invitrogen) and incubated at 4°C overnight, followed by mounting in Vectashield (Vector Laboratories) for imaging. The following antibodies were used: rabbit anti-GFP (1:1000, Invitrogen), mouse anti-brp (nc82) (1:50, DSHB), goat anti-rabbit IgG, and goat anti-mouse IgG (1:800, Alexa 488 or Alexa 633, respectively, Invitrogen). KCs were imaged using Leica TCS SP8 confocal microscope and LAS X software. Cells were counted using the point tool on Fiji 2.3.

## Quantitative PCR

Phenol-chloroform RNA extraction was performed with TRIzol (Ambion), and genomic DNA eliminated via DNase I (NEB). cDNA was generated using TaqMan RT Reagents (Applied Biosciences). RT-qPCR was performed using PowerUp SYBR Green Master Mix (Applied Biosciences) and an Applied Biosystems 7900HT Fast Real-Time PCR System. Samples were run in quadruplicate and raw Ct values extracted and averaged. Values for $\Delta Ct$ were exported to Prism/GraphPad for significance testing. Knockdown efficiency was calculated from $\Delta\Delta CT$ values representing the fold change relative to the housekeeping gene. The following primers were used: Cacophony: 5′-GCAAGGCGAAGCT GAGTTAC-3′ (Forward), 5′-AGGCGTTGACACCACAATTC-3′ (Reverse) and Rps20: 5′-TGTGGTGAG GGTTCCAAGAC-3′ (Forward), 5′-GACGATCTCAGAGGGCGAGT-3′ (Reverse).

## Quantification and statistical analysis

Data were compared using Wilcoxon rank-sum tests (two groups, nonparametric), ANOVA/Sidak (multiple comparisons, parametric) or Kruskal–Wallis/Dunn (multiple comparisons, nonparametric) tests. Box plots show graph the median as a line, the 1st and 3rd quartile enclosed in the box, and whiskers extending from the 10th to the 90th percentile.

## Acknowledgements

Fly stocks obtained from Krystyna Keleman and the Bloomington *Drosophila* Stock Center (NIH P40OD018537) were used in this study. The authors thank Yuexuan Li for the help in the development of the GRAB-ACh sensor, Brock Grill for helpful discussions, and Melissa Benilous for administrative assistance. Research support was provided by NIH R00MH092294 (S.M.T.), R01 NS097237 (S.M.T.), R01 NS114403 (S.M.T.), the Whitehall Foundation (S.M.T.), NIH R35NS097224 (R.L.D.), the Beijing Municipal Science & Technology Commission Z181100001318002 (Y.L.), the Beijing Brain Initiative of Beijing Municipal Science & Technology Commission Z181100001518004 (Y.L.), Guangdong Grant "Key Technologies for Treatment of Brain Disorders" 2018B030332001 (Y.L.), the General Program of National Natural Science Foundation of China projects 31671118, 31871087, and 31925017 (Y.L.), the NIH BRAIN Initiative NS103558 (Y.L.), grants from the Peking-Tsinghua Center for Life Sciences (Y.L.) and the State Key Laboratory of Membrane Biology at Peking University School of Life Sciences (Y.L.).

## Additional information

### Funding

| Funder | Grant reference number | Author |
| --- | --- | --- |
| National Institutes of Health | R01NS097237 | Seth M Tomchik |
| National Institutes of Health | R01NS114403 | Seth M Tomchik |
| National Institutes of Health | R00MH092294 | Seth M Tomchik |
| Whitehall Foundation | 2014-12-31 | Seth M Tomchik |
| National Institutes of Health | R35NS097224 | Ronald L Davis |

| Funder | Grant reference number | Author |
|---|---|---|
| Beijing Municipal Science & Technology Commission | Z181100001318002 | Yulong Li |
| Beijing Brain Initiative of Beijing Municipal Science & Technology Commission | Z181100001518004 | Yulong Li |
| Guangdong Grant "Key Technologies for Treatment of Brain Disorders" | 2018B030332001 | Yulong Li |
| General Program of National Natural Science Foundation of China Projects | 31671118 | Yulong Li |
| National Institutes of Health | NS103558 | Yulong Li |
| General Program of National Natural Science Foundation of China Projects | 31925017 | Yulong Li |
| General Program of National Natural Science Foundation of China Projects | 31871087 | Yulong Li |

The funders had no role in study design, data collection, and interpretation, or the decision to submit the work for publication.

## Author contributions

Aaron Stahl, Conceptualization, Data curation, Formal analysis, Investigation, Methodology, Validation, Visualization, Writing - original draft, Writing – review and editing; Nathaniel C Noyes, Data curation, Formal analysis, Investigation; Tamara Boto, Valentina Botero, Connor N Broyles, Formal analysis, Investigation; Miao Jing, Jianzhi Zeng, Methodology, Resources; Lanikea B King, Investigation; Yulong Li, Funding acquisition, Methodology, Resources, Writing – review and editing; Ronald L Davis, Data curation, Funding acquisition, Methodology, Project administration, Resources, Supervision, Validation, Writing – review and editing; Seth M Tomchik, Conceptualization, Data curation, Formal analysis, Funding acquisition, Methodology, Project administration, Resources, Software, Supervision, Validation, Visualization, Writing - original draft

## Author ORCIDs

Aaron Stahl (ID) http://orcid.org/0000-0003-3170-1101
Tamara Boto (ID) http://orcid.org/0000-0002-9974-3714
Valentina Botero (ID) http://orcid.org/0000-0002-9744-3929
Connor N Broyles (ID) http://orcid.org/0000-0002-2930-7343
Jianzhi Zeng (ID) http://orcid.org/0000-0002-5380-6281
Seth M Tomchik (ID) http://orcid.org/0000-0001-5686-0833

## Decision letter and Author response

Decision letter https://doi.org/10.7554/eLife.76712.sa1
Author response https://doi.org/10.7554/eLife.76712.sa2

# Additional files

## Supplementary files

• Transparent reporting form

## Data availability

Raw data have been deposited to Dryad under https://doi.org/10.5061/dryad.dfn2z353h.

The following dataset was generated:

| Author(s) | Year | Dataset title | Dataset URL | Database and Identifier |
|---|---|---|---|---|
| Tomchik SM | 2022 | Data from: Associative learning drives longitudinally-graded presynaptic plasticity of neurotransmitter release along axonal compartments | http://dx.doi.org/10.5061/dryad.dfn2z353h | Dryad Digital Repository, 10.5061/dryad.dfn2z353h |

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
