## [Editor Report]

This manuscript will be of interest to scientists working on learning and memory and synaptic plasticity. The study mostly uses an acetylcholine sensor in the fly brain to image activity, which is novel and helps to tie together previous studies reporting memory-induced changes in calcium transients. In particular, the study highlights the compartmentalised plasticity along Kenyon cell axon terminals, the main cell type of the insect mushroom body.

---

## [Decision Letter]

**Decision letter after peer review:**

[Editors’ note: the authors submitted for reconsideration following the decision after peer review. What follows is the decision letter after the first round of review.]

Thank you for submitting the paper "Associative learning drives longitudinally-graded presynaptic plasticity of neurotransmitter release along axonal compartments" for consideration by *eLife*. Your article has been reviewed by 3 peer reviewers, and the evaluation has been overseen by a Reviewing Editor and a Senior Editor. The reviewers have opted to remain anonymous.

Comments to the Authors:

We are sorry to say that, after consultation with the reviewers, we have decided to reject the current manuscript because of several substantial issues that were raised by the reviewers. Although we believe that these issues can be addressed experimentally and through text changes, we do not think that this substantial amount of work can be carried out within 2 months, the time *eLife* allows for revisions. As you will see, the reviewers were generally supportive of publication, but identified several key points such as (1) lack of behavioral experiments for the Cac knock-down experiments, (2) specific controls for some of the imaging experiments, (3) consideration of the role of dopaminergic neurons and (4) a lack of acknowledgment of the complexity of the mushroom body circuit and the literature that has addressed this previously. If the authors decide to address these issues, we are willing to consider a future manuscript, but it would be considered as a new submission.

*Reviewer #1:*

The paper aims to understand if and how axonal compartmentalization functions in the MB γ lobe in the context of olfactory learning. Further the authors focus on these process and find distinct molecular mechanisms underlying appetitive and aversive learning (cac and IP3). Lastly the authors focus on how this compartmentalization influences the downstream MBON function.

The paper is written well and straightforward in its approach and implications. The neuroanatomy, experimental details are clearly presented and I would rate this paper very high on a readability scale. The imaging and behavioral approaches are appropriate for the research question and address the limitations from a technical perspective. Results are presented well, but authors don't dwell on the results much before transitioning into another part of the question they seek to ask. This undermines the complexity of the MB circuits and an effort should be made to address dynamics of other lobes that underlie these behaviors. One issue is that even though the experiments work well together in supporting the results they fail to incorporate more complex dynamics of other lobes or consider the EM connectivity which might complicate the simplistic interpretations. The role of dopamine in this compartmental logic has not been directly addressed which potentially plays into this circuit.

This is impactful work as it addresses the pre-synaptic dynamics of MB with a focus on ACh release across compartments and how they differ between two forms of learning. The use of Grab Ach is a big highlight of the paper as this question could not have been asked without this tool. The authors also address the regulation of Ca^2+^ responses in the compartments that result in altered responses in valence-coding MBONs. Its also interesting to see the switch in γ2 and γ3 dynamics between appetitive and aversive learning. Lastly the match of Ca^2+^ responses in MBONs with the Ach compartmentalization highlights the behavioral relevance of axonal compartmentalization.

I like this paper as the experiments, results are clear and authors are very careful about interpreting the results. On a few occasions I feel that interpretations are over simplified and a focus on elaborating the complexities will be more helpful for the field.

Below are a few suggestions:

1) The use of Gal4 lines seems to be inconsistent between figures. The rationale for choosing a specific targeting tool should be made explicit.

2) The olfactory learning literature has many odor pairs and the one used here (ethyl butyrate and IAA) is not the most common pair. Do learning scores, neuroanatomy involved differ based on chosen odor pairs. Are learning rules generalizable across odor pairs.

3) Figure 1: Having some images from the rig set up would help in seeing how authors chose ROI's for the different compartments. The schematics are great but dont give a good sense for the real imaging resolution.

4) Figure 3: A schematic here like figure 1K would be great to align the reader. Also the n's here are half of figure 1. Is there a specific reason for that? How was power determined for these experiments.

5) Figure 4: It is not clear why γ2 is the only compartment shown here.

6) The compartmentalized pre-synaptic plasticity mapping/mirroring the MBON Ca++ response is a great result but without knocking down Ach receptor in these MBONs I am not sure a direct link between Ach release and MBON Ca^2+^ response cannot be made.

7) The above concern is somewhat addressed for γ3 as they mediate timing and MBON from γ3 are required for appetitive learning but is not addressed for other compartments and MBONs.

8) While, PAM and PPL1 neurons have been briefly discussed their role in modulating KC and MBON plasticity is not addressed experimentally which goes against what is know about this circuit. A more elaborate discussion would be beneficial. How do dopaminergic circuits associated with γ4 interact with this circuit?

Overall I find that Ach compartmentalization is well justified but the downstream circuit connectivity and plasticity seems unclear. Without addressing the role of Ach receptors it is difficult to see the causal relationship between KC Ach release and MBON activity.

*Reviewer #2:*

Stahl et al. investigate presynaptic Kenyon cell plasticity in the context of appetitive and aversive memories, using a genetically-encoded acetylcholine receptor-based sensor as primary read-out (instead of calcium indicators used in several previous studies). Therefore, the authors investigate one step downstream of the classically conducted calcium (or cAMP, etc.) imaging experiments – the level of neurotransmitter release. Likewise, this is one step upstream of the other widely-used readout of the postsynaptic MBON activity (or dopaminergic neurons). The authors investigate contributions of CS+ and/or CS- plasticity throughout the compartments of the γ lobe. All in all, this manuscript confirms several previous studies that investigated individual compartment plasticity (often plasticity is measured at the level of the MBONs and therefore in single compartments), and taken together with other recent publications, this study can be seen as a valuable compendium (as it addresses appetitive and aversive short-term memory protocols in the context of several compartments). As the authors address an important step between presynaptic and postsynaptic calcium transients, their work actually confirms that conclusions deduced from changed calcium transients correlate to neurotransmitter release, as expected. This study also addresses differential effects on the CS+ and the CS-, which is important, as previous studies often concentrated (partially for technical reasons) on either the CS+ specifically, or the relative changes of CS+ to CS- only. The authors further investigate the role of the active zone-localized voltage gated calcium channel cacophony in memory-related plasticity – this part needs to be strengthened by additional controls.

1. As it stands, I am not sure what to read from the Cac knock-down experiments for several reasons. First, it is entirely unclear how efficient the knock-down works. Would the amount of Cac molecules be reduced from e.g., 20 to 15 per active zone, or less, or more (the authors should quantify cacophony following knock-down)? Second, if baseline transmission is not affected (which is remarkable to a certain degree – the authors do mention potential 'residual' Cac expression in the discussion), what about e.g., high frequency odor puff protocols (again the authors mention plasticity protocols in the discussion that may help tying the observed effects to Cac). Can the authors perform additional manipulations that should phenocopy the reduction of Cac (e.g., reduction of Brp via RNAi)? Can the phenotype be modulated by titrating extracellular calcium concentrations? Would basal neurotransmission be affected when extending the time at 32{degree sign}C (the range of 4-10 days could also be quite large, depending on the turnover time for cacophony). Third, can the authors exclude that the observed effects were due to dendritic functions of Cac?

The authors need to perform additional experiments in order to manifest this important claim.

2. What is the behavioral consequence of Cac or IP3R knock-down in Kenyon cells? Is memory impaired?

3. The authors actually do not demonstrate that the acetylcholine measured is released from MB neurons (Kenyon cells) and do not exclude the possibility that the measured acetylcholine could derive from an unknown other source. The authors could address this issue by Kenyon cell-specific knock-down of VAChT or ChaT. Alternatively, acute or chronic block of KCs, while measuring sensor activation, could be an option. I am puzzled by the wording in line 80: is 'putative' meant as 'generally accepted' or 'likely'? I would strongly recommend using a less ambiguous word in this context. If not generally accepted (if so, please do explain to the reader why), this would somewhat compromise the basis of this study.

4. Figure 1 and others: Why are only the means compared statistically? The pre to post in C and E should be analysed using paired statistics – as in Figure 3E/F or S4 for example. In Figure 1: are any changes detected within flies? Figure S1 suggest that this is not the case – the change would only become apparent between conditions? Although the authors have already done a good job in illustrating their protocols, data display should be more uniform potential discrepancies should be spelt out, and claims should be adjusted accordingly (e.g., summary in K).

5. Line 189: why a trend towards decrease? Figure S4 suggests a statistically significant change.

6. Absolute conditioning protocols: Is the comparison of the 'no CS-' and 'no CS+' really suited to claim that 'no CS-' leads to decreased acetylcholine release? It appears that this would only be consistent if the 'no CS-' group was significantly different from the 'no US' group. Please clarify putative discrepancies. In this context, please also discuss the results in S5 in more detail.

7. Figure S4 shows

a. gamma1: CS+ potentiation (not seen in Figure S1?), not apparent in Cac RNAi

b. gamma2: no potentiation for the CS+ (unlike what is reported in Figure 1), yet, consistently a depression for the CS-

c. gamma3: no potentiation of CS+, yet a depression of CS-, both inconsistent with Figure 1

d. Several forms of plasticity, also to the non-trained odor in controls and Cac RNAi

Please clarify and discuss the individual findings.

8. Line 467: It is first stated that odor was presented continuously for 30s. In the next sentence the authors write that six 1-s odor pulses with a 5s inter-pulse interval were given. Please clarify. Also, the odor responses appear to typically last for about 5-6s. Obviously, this is in parts based on the kinetics of the reporter. However, would the authors not expect that continuous odor application will lead to desensitization at the level of the olfactory receptor neurons?

9. Apart from showing that postsynaptic calcium signals are altered in MBONs the authors should consider showing whether the observed effects seen with the acetylcholine indicator can also be seen when expressed in MBONs or DANs (and exclude the possibility that the potentiation/depression could be specific to KC-DAN or KC-KC synapses).

*Reviewer #3:*

This study shows learning-induced compartmentalized plasticity in Kenyon cells (KCs) of the mushroom body γ lobe using a genetically encoded acetylcholine sensor. The authors made several major discoveries that the plasticity is bidirectional and depends on memory valence. Interestingly, the compartments along the axon terminals of the KCs undergo spatial 'gradient' of plasticity. Knock-down of the genes that differentially regulate intracellular calcium induced distinct effects on learning-induced plasticity. These results explain, at least partly, that learning-induced plasticity at KC-MBON synapses takes place on the presynaptic side. The experiments and analysis are overall thorough, and conclusions are generally supported by the results. In the following, I list some flaws to be addressed that concern necessary controls, more careful interpretation of the statistics and results, and the choice of the cac RNAi strain.

- Odor-only control is imperfect to formally distinguish if the plasticity arises from CS+ and/or CS- changes. The effect of sugar or shock presentation per se must be measured to make this distinction. Alternatively, the authors can examine how much of the 'plasticity' is explained by unpaired presentations of CS and US.

- Conditioning effects at some compartments are on the edge of statistical significance and look inconsistent among different experimental series (e.g. Figure 1F-J vs. Figure 3F). The authors need to be cautious to interpret these statistical results. For example, knock-down of cac impaired CS- depression in y2 and y3 as well (cf. Figure 1G, H, and K), and this is contradictory to their conclusion that cac is required only for potentiation. Similarly, itpr seem to globally affect presynaptic activities.

- Some of the major findings here have been consistent with previous papers with different measurements (e.g. Bilz et al. 2020 for Figure 2). On the other hand, this study presents results not coherent with existing studies. For example, previous papers of the authors found rather homogeneous calcium increase (Boto 2014, Louis 2018), while plasticity in this study is graded along the y lobe axons and opposite in appetitive and aversive conditioning (Figure 1, 2). MBON-y3 has been reported to be dispensable (Aso 2014 and Ichinose 2021) unlike the results in Figure 6E. The authors need to provide discussion that explain the coherence/apparent inconsistencies.

- The choice of this particular cac RNAi strain is not clear, besides no control experiment for off-target effects. This is technically important, since voltage-gated calcium channels must be required more than learning-dependent potentiation. Therefore, it is blunt to conclude:

"synaptic exocytosis remained intact (L191)" upon cac knock-down;

"presynaptic potentiation, but not depression, requires the voltage-sensitive CaV2 Ca^2+^ channel cacophony across the MB compartments (L198)";

"Overall, these data demonstrate that Cac is not required for learning-induced depression of ACh release (L207)".

These conclusions need to be moderated.

Also in this context, mentioning a recent paper by Hidalgo et al. (Neurobiology of Disease, 2021) that analyzed the effect of cac-KD in KCs on aversive learning would be complementary to the physiological finding of this study.

- Compartmentalized and bidirectional plasticity in the MBONs has been repeatedly reported (e.g. Cohn, 2015; Owald, 2015). What is the intention to revisit this (Figure 5)? Also, this is essentially a detour from their main argument about the presynaptic mechanism. Consider revising the Figure or moving it to the figure supplement for readability. Similarly, it's not clear to me what Figure 6 adds to the overall conclusion.

- (Figure 6E) Did the authors examine MBON blocking in discriminative training? This needs to be clearly described. If the authors claim that the y3 compartment is important in discriminative training, they would need to show no defects in single-odor conditioning as in Figure 6A.

- Selection of ROIs for different MBONs in Figure 5 need stronger rationale. Why do the authors choose neurites for some compartments and not for others? I guess there is transformation of calcium signals in different parts of MBONs.

- Describe how the authors defined compartments in the g-KCs?

- I'm not totally sure how the results in Figure 6 support "temporal comparison (L275)". Either elaborate this conclusion more or provide an alternative interpretation.

- The quantified compartment for Figure 4A-D should be clearly stated in the text and legend, instead of simply showing y2 in 4E and let us guess.

- Is "post-conditioning odor contrast" appropriate for explaining itpr effects given the CS+ depression is seemingly exaggerated (Figure 4)?

- L256 Sentence and/or figure citation are somewhat inconsistent.

[Editors’ note: further revisions were suggested prior to acceptance, as described below.]

Thank you for resubmitting your work entitled "Associative learning drives longitudinally-graded presynaptic plasticity of neurotransmitter release along axonal compartments" for further consideration by *eLife*. Your revised article has been evaluated by Gary Westbrook (Senior Editor) and a Reviewing Editor. The manuscript has been improved but there are some remaining issues that need to be addressed, as outlined below:

The reviewers were overall happy with the revisions, but both raised a few points that need to be addressed before a final decision can be reached. Please pay attention in particular to the points 1. and 2. raised by reviewer 2. We believe these can be addressed by editing the text and toning down relevant statements.

*Reviewer #1:*

The authors have done a good job with this revised manuscript. They have addressed my previous concerns either by adding new experiments or clarifying passages in the text. Especially the role of Cac in appetitive learning and the more clear description of their analysis pipeline are convincing. Some experiments suggested, were technical not achievable with the presented techniques, and pursuing alternate methods would exceed the scope of this study.

*Reviewer #2:*

I've read the revised manuscript, and found it much improved. However, some of my concerns were not addressed. The authors need to clarify the following points before publication.

1. The odor-only control accounts for the effect of non-associative olfactory adaptation. But only that. The authors should be aware that other types of non-associative plasticity taking place during learning; e.g. reduced odor acuity by the previous exposure to electric shock (Preat, J Neurosci, 1998). Thus, inclusion of the US-only and/or unpaired training controls is mandatory to formally determine the absolute plasticity effect (e.g. potentiation of CS+ or depression CS- are indistinguishable; Figures 1L and 2K). Otherwise, they should moderate these claims, and focus only on the relative effects (i.e. "CS+:CS-" in the same figures)..

2. As demonstrated by the authors, Cac-KD reduces only ~29% of total Cac RNA level, suggesting that large body of Cac expression is unaffected. It is quite questionable that Cac is not required for learning-induced depression of Ach release (P10, L234) based on this partial down-regulation.

3. Somehow related to the previous point. Is it reasonable to conclude that Cac is required for learning-dependent potentiation? There is potentiation of CS- odor response in the γ 5 compartment after appetitive learning (Figure 3- sup1E)?

---

## [Author Response]

[Editors’ note: the authors resubmitted a revised version of the paper for consideration. What follows is the authors’ response to the first round of review.]

Reviewer #1:[…]I like this paper as the experiments, results are clear and authors are very careful about interpreting the results. On a few occasions I feel that interpretations are over simplified and a focus on elaborating the complexities will be more helpful for the field.

Thank you for the constructive feedback – we attempted to balance a clean focus on the results at hand with discussion of the many other impressive recent studies illuminating structural and functional aspects of the MB & associated circuitry. As suggested, in the present version, we have further elaborated on the complexities of these circuits to provide the readers with a more comprehensive picture. This is woven throughout the manuscript, particularly the discussion (e.g., pages 22‐23).

Below are a few suggestions:1) The use of Gal4 lines seems to be inconsistent between figures. The rationale for choosing a specific targeting tool should be made explicit.

It has now been made explicit in the manuscript text (in the Methods, page 24, and main text, pages 5,9). We used two Gal4 lines that express in KCs (including γ KCs): R13F02 and 238Y. Initial ACh imaging experiments were carried out using 238Y to drive GRAB‐ACh expression; this provides maximal KC coverage, labeling 1898/~2000 KCs (Aso et al., 2009, J Neurogenet 23:156‐72). However, 238Y also labels a relatively large number of other cell types outside the MB (ibid). We often observe lethality when driving various RNAi lines with this driver, likely due to the non‐KC expression in the driver. In addition, the driver produces developmental defects when expressing various RNAi lines throughout development, reinforcing the well‐known potential for such issues when expression is not restricted to adulthood. Therefore, to mitigate these risks the molecular experiments, we switched to a more KC‐selective driver, R13F02‐Gal4, and restricted the expression of the RNAi (and GRAB‐ACh) to adulthood via Gal80^ts^. The number of KCs labeled by R13F02‐Gal4 had not been documented (to our knowledge). As this is a relevant parameter for the comparison of results across drivers, we labeled and counted them, finding that the driver labels 745 ± 47 KCs; this is now included in the manuscript. Overall, the use of these two drivers/strategies provides complementary data sets, with 238Y providing insight into the plasticity across KCs and compartments, with the broadest possible KC coverage, and R13F02 providing more selective KC manipulations in the molecular experiments.

2) The olfactory learning literature has many odor pairs and the one used here (ethyl butyrate and IAA) is not the most common pair. Do learning scores, neuroanatomy involved differ based on chosen odor pairs. Are learning rules generalizable across odor pairs.

Indeed, many odor pairs have been used successfully, including various pairs and binary mixtures of 3octanol, 4‐methylcyclohexanol (Mch), benzaldehyde, 1‐octen‐3‐ol (Oct), pentyl acetate, butyl acetate, ethyl butyrate (EB), and isoamyl acetate (IA) (e.g., Barth et al., 2014 J Neurosci 34: 1819‐1837). We have previously used EB and IA as an odor pair for behavior olfactory associative conditioning (Boto et al., 2019 Cell Rep 27: 2014–2021) and for olfactory stimuli in imaging experiments (Boto et al., 2014 24: 822‐831; Louis et al., 2018 PNAS 115: E448‐E457). Performance indexes in behavioral conditioning experiments are equivalent to another odor pair (Mch vs. Oct) (Boto et al., 2019), and we see no substantive difference in odor‐evoked responses across KCs. Learning rules generalize across odors. Thus, by all available evidence, this odor pair is equally representative of odor space to any other.

3) Figure 1: Having some images from the rig set up would help in seeing how authors chose ROI's for the different compartments. The schematics are great but dont give a good sense for the real imaging resolution.

Agreed – an image is now included (Figure 1B).

4) Figure 3: A schematic here like figure 1K would be great to align the reader.

We appreciate the suggestion ‐ a schematic diagram is now included for figure 3, as well as figures 2 and 4.

Also the n's here are half of figure 1. Is there a specific reason for that? How was power determined for these experiments.

Sample sizes for aversive conditioning were based on prior published studies that detected positive effects (e.g., Louis et al., 2018). For appetitive conditioning/imaging experiments, we ran a priori power analysis to determine appropriate sample sizes using G*Power 3.1, with the following criteria: two‐tailed, α = 0.05, Power (1‐β) = 0.8, moderate effect size (α = 0.5), arriving at a total n ≥ 26 (we treated this as the minimum and ran 27).

5) Figure 4: It is not clear why γ2 is the only compartment shown here.

In figure 4, we focused on one compartment (in the main figure) to cleanly convey the conclusions in a representative compartment. Comparing Cac and IP3 RNAi, showing pre/post quantifications for CS+, CS‐, EB, and IA and box plots for one compartment introduces a relatively large number of conditions for the reader to track. Multiplying this five times over made for an unwieldy figure (we tried in earlier draft versions). For that level of detail, we direct the readers to Figure 4 – Supplement 1, which includes all compartments (stretching to 30 panels). Supplemental Figures appear side‐by‐side with a single click to toggle the view on the *eLife* web site upon publication, so the readers will be able to readily view all of the data. Nonetheless, to add spatial detail to the main Figure 4, we have now included a diagram of the plasticity effects across compartments (Figure 4, panels C,F,I).

6) The compartmentalized pre-synaptic plasticity mapping/mirroring the MBON Ca++ response is a great result but without knocking down Ach receptor in these MBONs I am not sure a direct link between Ach release and MBON Ca^2+^ response cannot be made.

These experiments verify that plasticity drives responses in MBONs in the same direction as the KCs. Plasticity in MBONs likely reflects both pre‐ and post‐synaptic processes (e.g., Pribbenow et al., 2021, bioRxiv doi: 10.1101/2021.07.01.450776) (we now discuss this aspect further in the manuscript). Knockdown of the postsynaptic ACh receptors in this context would remove the input to MBONs but not isolate the effects of ACh release. The best way to do so is to monitor ACh release and manipulate presynaptic processes (e.g., presynaptic Cac Cav2 channels), which is the approach we selected. We have now further enhance the rigor of these experiments by blocking synaptic release from the KCs with *Shibire*^ts^ (Figure 4 – Supplement 3), which significantly inhibited the odor‐evoked ACh signal.

7) The above concern is somewhat addressed for γ3 as they mediate timing and MBON from γ3 are required for appetitive learning but is not addressed for other compartments and MBONs.

We focused on the γ3 MBONs due to their novelty in terms of [presynaptic] CS‐ effects and the relative paucity of studies examining their roles in behavioral conditioning. In this updated manuscript, we have removed the timing experiments, focusing on the fundamental requirement for the γ3 MBONs in appetitive conditioning (γ2α’1 MBONs were also tested behaviorally).

8) While, PAM and PPL1 neurons have been briefly discussed their role in modulating KC and MBON plasticity is not addressed experimentally which goes against what is know about this circuit. A more elaborate discussion would be beneficial. How do dopaminergic circuits associated with γ4 interact with this circuit?

We have incorporated more discussion of these neurons in bidirectional plasticity in the discussion.

Overall I find that Ach compartmentalization is well justified but the downstream circuit connectivity and plasticity seems unclear. Without addressing the role of Ach receptors it is difficult to see the causal relationship between KC Ach release and MBON activity.

Our focus in this manuscript is on presynaptic plasticity in ACh release in KCs. As the reviewer notes, there is plasticity across multiple loci, including the MBONs. We added citations to additional studies to incorporate this point. In addition, we have now bolstered the experiments testing the presynaptic release mechanisms in several ways. We note that postsynaptic plasticity is being addressed in a complementary study from another group (Pribbenow et al., 2021).

Reviewer #2:Stahl et al. investigate presynaptic Kenyon cell plasticity in the context of appetitive and aversive memories, using a genetically-encoded acetylcholine receptor-based sensor as primary read-out (instead of calcium indicators used in several previous studies). Therefore, the authors investigate one step downstream of the classically conducted calcium (or cAMP, etc.) imaging experiments – the level of neurotransmitter release. Likewise, this is one step upstream of the other widely-used readout of the postsynaptic MBON activity (or dopaminergic neurons). The authors investigate contributions of CS+ and/or CS- plasticity throughout the compartments of the γ lobe. All in all, this manuscript confirms several previous studies that investigated individual compartment plasticity (often plasticity is measured at the level of the MBONs and therefore in single compartments), and taken together with other recent publications, this study can be seen as a valuable compendium (as it addresses appetitive and aversive short-term memory protocols in the context of several compartments). As the authors address an important step between presynaptic and postsynaptic calcium transients, their work actually confirms that conclusions deduced from changed calcium transients correlate to neurotransmitter release, as expected. This study also addresses differential effects on the CS+ and the CS-, which is important, as previous studies often concentrated (partially for technical reasons) on either the CS+ specifically, or the relative changes of CS+ to CS- only. The authors further investigate the role of the active zone-localized voltage gated calcium channel cacophony in memory-related plasticity – this part needs to be strengthened by additional controls.1. As it stands, I am not sure what to read from the Cac knock-down experiments for several reasons. First, it is entirely unclear how efficient the knock-down works. Would the amount of Cac molecules be reduced from e.g., 20 to 15 per active zone, or less, or more (the authors should quantify cacophony following knock-down)? Second, if baseline transmission is not affected (which is remarkable to a certain degree – the authors do mention potential 'residual' Cac expression in the discussion), what about e.g., high frequency odor puff protocols (again the authors mention plasticity protocols in the discussion that may help tying the observed effects to Cac).

The reviewer raises valid points about the need for more detail on the RNAi strategy and further characterization of the knockdown. The goal was to knock down Cac enough to test its role in plasticity, but not completely disrupt synaptic transmission. This is similar in approach to experiments previously carried out at the neuromuscular junction (Frank et al., 2006, Neuron 52:663‐77; Muller et al., 2012, Curr Biol 22:1102‐8). We chose an RNAi line that drives moderate knockdown (Brusich et al., 2015, Front. Cell. Neurosci 9:107) and expressed it conditionally under Gal80^ts^ control. These details are now discussed more clearly in the manuscript. In addition, as requested, we quantified efficiency of the knockdown with quantitative PCR, expressing the RNAi pan‐neuronally and using the same Gal80ts/induction protocol. This revealed a ~29% reduction in Cac mRNA levels. Further details about the strategy and new experiment have been added to the manuscript in the main text and Methods.

Overall, the data demonstrate that Cac is necessary for the potentiation in proximal γ lobe compartments, identifying a key mechanism of learning‐induced presynaptic plasticity in the KCs. This is consistent with the role that Cac plays at the neuromuscular junction, where Cac levels are positively correlated with synaptic potentiation (Gratz et al., 2019; J Neurosci). Furthermore, presynaptic homeostatic potentiation at the neuromuscular junction is dependent on Cac and Ca^2+^ levels (Frank et al., 2006, Neuron 52:663‐77; Muller et al., 2012, Curr Biol 22:1102‐8). Finally, Ca_V_2.1 levels at the active zone are dynamically regulated and positively drive synaptic strength in mammals as well (Lubbert et al., 2018, Neuron 101:260‐273).

Can the authors perform additional manipulations that should phenocopy the reduction of Cac (e.g., reduction of Brp via RNAi)?

We have not attempted Brp knockdown in this study, as it would likely impair synaptic transmission across multiple compartments in a non‐valence‐specific manner. In addition, it would not provide direct insight into plasticity mechanisms and does not have (to our knowledge) a validated RNAi line allowing mild knockdown.

Can the phenotype be modulated by titrating extracellular calcium concentrations?

This is a conceptually interesting idea, but would not be feasible in vivo, where olfactory responses must be allowed to propagate unperturbed across multiple neurons/synapses to reach the KCs (from olfactory receptor neurons to projection neurons to KCs).

Would basal neurotransmission be affected when extending the time at 32{degree sign}C (the range of 4-10 days could also be quite large, depending on the turnover time for cacophony).

Basal neurotransmission could be affected if we were to increase the expression level of the RNAi (through various means), though doing so would be counterproductive for our experimental purpose.

Third, can the authors exclude that the observed effects were due to dendritic functions of Cac?

The axonal localization of Cac function is supported by its [axon] compartment‐specific effects on potentiation. Dendritic effects propagating into the axons would be distributed across multiple compartments (e.g., Boto et al., 2019, Cell Rep 27: 2014‐2021).

The authors need to perform additional experiments in order to manifest this important claim.

Multiple new experiments have added to the depth and rigor of the analysis. Overall, these experiments succeeded in allowing us to pinpoint Cac as a molecule that is important for regulating presynaptic plasticity and provided a molecular mechanism to manipulate/differentiate the plasticity occurring across the spatial compartments. The data demonstrated that Cac is necessary for the potentiation in proximal γ lobe compartments, identifying a key mechanism of learning‐induced presynaptic plasticity in the mushroom body. This is consistent with the role that Cac plays at the neuromuscular junction, where Cac levels are positively correlated with synaptic potentiation (Gratz et al., 2019; J Neurosci). Furthermore, presynaptic homeostatic potentiation at the neuromuscular junction is dependent on Cac and Ca^2+^ levels (Frank et al., 2006, Neuron 52:663‐77; Muller et al., 2012, Curr Biol 22:1102‐8). Finally, Ca_V_2.1 levels at the active zone are also dynamically regulated and positively drive synaptic strength in mammals (Lubbert et al., 2018, Neuron 101:260‐273).

2. What is the behavioral consequence of Cac or IP3R knock-down in Kenyon cells? Is memory impaired?

To test this, we knocked down Cac and carried out appetitive conditioning, finding that short‐term memory was indeed impaired (Figure 3H).

3. The authors actually do not demonstrate that the acetylcholine measured is released from MB neurons (Kenyon cells) and do not exclude the possibility that the measured acetylcholine could derive from an unknown other source. The authors could address this issue by Kenyon cell-specific knock-down of VAChT or ChaT. Alternatively, acute or chronic block of KCs, while measuring sensor activation, could be an option.

We have now addressed this with the latter experiment that the reviewer suggested – blocking KC output while monitoring ACh release with GRAB‐ACh. The ACh signal was inhibited (Figure 4 – Supplement 3). Some signal was still observed, which likely emanates from the KCs that are not labeled by the driver and continue to respond to odors and release ACh (R13F02 labels ~746/2000 KCs). While the ACh signal imaged in the MB is clearly dominated by KCs, we do not exclude that there could be some contribution from non‐KCs, and added that caveat to the discussion.

I am puzzled by the wording in line 80: is 'putative' meant as 'generally accepted' or 'likely'? I would strongly recommend using a less ambiguous word in this context. If not generally accepted (if so, please do explain to the reader why), this would somewhat compromise the basis of this study.

Agreed – ACh is the described neurotransmitter from KCs. We were trying to convey that, without ruling out that some hypothetical future study could report cotransmission of some other neurotransmitter(s) (as has occurred in some other cell types), but the language was inaccurate on its face. This has been clarified. The sentence now reads “…synaptic release of the KC neurotransmitter…”.

4. Figure 1 and others: Why are only the means compared statistically? The pre to post in C and E should be analysed using paired statistics – as in Figure 3E/F or S4 for example. In Figure 1: are any changes detected within flies? Figure S1 suggest that this is not the case – the change would only become apparent between conditions? Although the authors have already done a good job in illustrating their protocols, data display should be more uniform potential discrepancies should be spelt out, and claims should be adjusted accordingly (e.g., summary in K).

We run all these comparisons on all data sets (Figure 1 – Supplement 1 and others). The key metric to detect potential/depression is the comparison of the Δ(post/pre) between each experimental group and its respective odor‐only control group (e.g., CS+ vs. EB). This is because olfactory conditioning generates some alteration/adaptation of the olfactory responses simply as a function of exposing the animal to the odor. The post vs. pre comparison sometimes accurately reveals such differences, particularly if they are large (e.g., in Ca^2+^ imaging experiments at the whole‐cell level, and, as the reviewer notes, in our subcellular data as well, in places). However, considering only the post vs. pre comparison would produce false negatives in some compartments that potentiate, as well as false positives in some compartments that do not. Therefore, we rely on the comparison of Δ(post/pre) between each experimental group and its respective odor‐only control as the primary metric. As noted above, we also ran the post vs. pre comparisons and report significant differences where they were found (and trends where they are important and informative).

5. Line 189: why a trend towards decrease? Figure S4 suggests a statistically significant change.

Following the rationale above, we only counted a change as significant if it differed from the odor-only control in terms of Δ(post/pre). The reviewer is correct – it is conservative here. This is often the case, as it requires that there be both a difference in Δ(post/pre) and that the difference be larger than any Δ(post/pre) in the odor‐only control.

6. Absolute conditioning protocols: Is the comparison of the 'no CS-' and 'no CS+' really suited to claim that 'no CS-' leads to decreased acetylcholine release? It appears that this would only be consistent if the 'no CS-' group was significantly different from the 'no US' group. Please clarify putative discrepancies. In this context, please also discuss the results in S5 in more detail.

Upon consideration of the reviewers’ comments and the overall theme/flow of the manuscript, we have removed this experiment from the manuscript. We intend to follow up on this result more thoroughly in a future study. For now, we focus on the conclusion that the γ3 compartment exhibits alterations in ACh release, and that its downstream MBONs are required for appetitive conditioning.

7. Figure S4 showsa. gamma1: CS+ potentiation (not seen in Figure S1?), not apparent in Cac RNAib. gamma2: no potentiation for the CS+ (unlike what is reported in Figure 1), yet, consistently a depression for the CS-c. gamma3: no potentiation of CS+, yet a depression of CS-, both inconsistent with Figure 1d. Several forms of plasticity, also to the non-trained odor in controls and Cac RNAiPlease clarify and discuss the individual findings.

There are several differences between the genotypes in these experiments: (1) Gal4 driver (238Y vs. [sparser] R13F02), (2) inclusion of Gal80^ts^, (3) temperature shifts for Gal80^ts^ induction. One or more of these differences may have slightly altered the plasticity effects in controls, though the overall conclusions are consistent: appetitive conditioning potentiated the relative CS+/CS‐ responses in the proximal γ compartments, while generating a trend toward decreasing responses in the distal compartments. This is now discussed in more detail in the manuscript.

8. Line 467: It is first stated that odor was presented continuously for 30s. In the next sentence the authors write that six 1-s odor pulses with a 5s inter-pulse interval were given. Please clarify. Also, the odor responses appear to typically last for about 5-6s. Obviously, this is in parts based on the kinetics of the reporter. However, would the authors not expect that continuous odor application will lead to desensitization at the level of the olfactory receptor neurons?

This was an error in the protocol description that has been corrected. For ACh imaging experiments, short odor pulses were used to avoid desensitization of the GRAB‐ACh sensor. The reviewer is correct, the responses last longer than the odor pulse, likely due to the kinetics of the reporter. Continuous odor application was used only in behavioral experiments, where this is a standard approach (calibration of the odor concentrations in the T‐maze to balance the naïve distribution accounts for any imbalance in desensitization between odors).

9. Apart from showing that postsynaptic calcium signals are altered in MBONs the authors should consider showing whether the observed effects seen with the acetylcholine indicator can also be seen when expressed in MBONs or DANs (and exclude the possibility that the potentiation/depression could be specific to KC-DAN or KC-KC synapses).

Expression levels of the GRAB‐ACh reporter are too low when expressed postsynaptically in the MBONs that we have tested so far.

Reviewer #3:This study shows learning-induced compartmentalized plasticity in Kenyon cells (KCs) of the mushroom body γ lobe using a genetically encoded acetylcholine sensor. The authors made several major discoveries that the plasticity is bidirectional and depends on memory valence. Interestingly, the compartments along the axon terminals of the KCs undergo spatial 'gradient' of plasticity. Knock-down of the genes that differentially regulate intracellular calcium induced distinct effects on learning-induced plasticity. These results explain, at least partly, that learning-induced plasticity at KC-MBON synapses takes place on the presynaptic side. The experiments and analysis are overall thorough, and conclusions are generally supported by the results. In the following, I list some flaws to be addressed that concern necessary controls, more careful interpretation of the statistics and results, and the choice of the cac RNAi strain.- Odor-only control is imperfect to formally distinguish if the plasticity arises from CS+ and/or CS- changes. The effect of sugar or shock presentation per se must be measured to make this distinction. Alternatively, the authors can examine how much of the 'plasticity' is explained by unpaired presentations of CS and US.

The odor‐only control accounts for the effect of non‐associative olfactory adaptation. Comparison of the CS+ and CS‐ in experimental groups reveal the associative changes in ACh release.

- Conditioning effects at some compartments are on the edge of statistical significance and look inconsistent among different experimental series (e.g. Figure 1F-J vs. Figure 3F). The authors need to be cautious to interpret these statistical results. For example, knock-down of cac impaired CS- depression in y2 and y3 as well (cf. Figure 1G, H, and K), and this is contradictory to their conclusion that cac is required only for potentiation. Similarly, itpr seem to globally affect presynaptic activities.

Cac knockdown did not affect CS‐ depression: CS‐ depression was not observed in controls for this experiment. There are several differences between the control genotypes in these experiments: (1) Gal4 driver (238Y vs. [sparser] R13F02), (2) inclusion of Gal80^ts^, (3) temperature shifts for Gal80^ts^ induction. One or more of these differences may have slightly altered the plasticity effects in controls, though the overall conclusions are consistent: appetitive conditioning potentiated the relative CS+/CS‐ responses in the proximal γ compartments, while generating a trend toward decreasing responses in the distal compartments.

- Some of the major findings here have been consistent with previous papers with different measurements (e.g. Bilz et al. 2020 for Figure 2). On the other hand, this study presents results not coherent with existing studies. For example, previous papers of the authors found rather homogeneous calcium increase (Boto 2014, Louis 2018), while plasticity in this study is graded along the y lobe axons and opposite in appetitive and aversive conditioning (Figure 1, 2).

Indeed, this is an important finding here – Ca^2+^ in the presynaptic terminal is often, but not always, correlated with the directionality and magnitude of plasticity in ACh release across compartments. One inspiration for the present study was the independent lines of evidence suggesting presynaptic localization of plasticity associated with aversive learning (e.g., requirement for cAMP signaling molecules in KCs [e.g., Zars et al., 2000]) combined with our previous studies suggesting no net changes in odor‐evoked Ca^2+^ responses across MB γ compartments following aversive conditioning (at short‐term time points). This is discussed in the Introduction (lines 63‐67). These observations are not discrepancies, but rather reflect the different approaches used across the prior studies; the present data represent a step toward resolving these apparent discrepancies.

MBON-y3 has been reported to be dispensable (Aso 2014 and Ichinose 2021) unlike the results in Figure 6E. The authors need to provide discussion that explain the coherence/apparent inconsistencies.

Aso et al. 2014 tested 2hr appetitive memory (not STM), while silencing a subset of MBONs, which did not include the γ3 MBONs (their Figure 7B). Among the candidate GABAergic MBONs listed in the figure legend, they only tested the γ1pedc MBON in this particular assay. Ichinose et al. 2021 blocked MBONs in individual phases and tested at 1 hr, while we blocked across all phases and examined immediate memory. One of the major conclusions of their study was that multiple MBONs are involved in modulating reward memory. Given the differences in the approaches between the studies, we do not interpret the present results as necessarily inconsistent. Nonetheless, the paragraph describing the γ3 MBON behavioral data in the discussion has been edited to discuss these points.

- The choice of this particular cac RNAi strain is not clear, besides no control experiment for off-target effects. This is technically important, since voltage-gated calcium channels must be required more than learning-dependent potentiation. Therefore, it is blunt to conclude:"synaptic exocytosis remained intact (L191)" upon cac knock-down;"presynaptic potentiation, but not depression, requires the voltage-sensitive CaV2 Ca^2+^ channel cacophony across the MB compartments (L198)";"Overall, these data demonstrate that Cac is not required for learning-induced depression of ACh release (L207)".These conclusions need to be moderated.

The Cac line was chosen due to its moderate knockdown when driven without dcr‐2 (Brusich et al., 2015, Front. Cell. Neurosci 9:107). We have now included quantitative PCR demonstrating the efficacy of the RNAi line and confirming that we achieved the intended moderate knockdown effect. While we cannot rule out off‐target effects, we have confirmed that the manipulation produced the intended on‐target effect at the approximate desired level. Therefore, these statements represent conclusions that are supported by the data.

Also in this context, mentioning a recent paper by Hidalgo et al. (Neurobiology of Disease, 2021) that analyzed the effect of cac-KD in KCs on aversive learning would be complementary to the physiological finding of this study.

Agreed. We have now cited the Hidalgo et al. paper and run complementary behavioral tests using appetitive conditioning. Our results are consistent with the interpretation that Cac knockdown in KCs impacts appetitive learning, and the inclusion of these new data bolster the findings of the study.

- Compartmentalized and bidirectional plasticity in the MBONs has been repeatedly reported (e.g. Cohn, 2015; Owald, 2015). What is the intention to revisit this (Figure 5)? Also, this is essentially a detour from their main argument about the presynaptic mechanism. Consider revising the Figure or moving it to the figure supplement for readability.

The present experiments perform several functions: (1) demonstrating that plasticity is observed on the opposite side of the synapse with the same set of conditions that we used to test presynaptic plasticity, (2) testing plasticity following appetitive conditioning at the same time points we measured ACh release, (3) testing the role of the γ2α’1 and γ3 MBONs, which have not been well studied in the context of appetitive conditioning. Without these data, readers would be left to integrate conclusions across studies, assume that conditions were similar enough to compare across studies/labs, and extrapolate across multiple MBONs. While these are reasonable assumptions, they should be tested experimentally. To highlight the more novel findings, we moved the data from the γ1pedc and γ5β’2a MBONs to the supplement.

Similarly, it's not clear to me what Figure 6 adds to the overall conclusion.

In response to the reviewer’s inquiry, we reconsidered this data set and largely agree. We retained the experiment showing the requirement for the γ3 MBONs for normal appetitive short‐term conditioning (which shed light on the potential importance of plasticity in ACh release in the γ3 compartment), while eliminating the more tangential single‐odor temporal analyses.

- (Figure 6E) Did the authors examine MBON blocking in discriminative training? This needs to be clearly described. If the authors claim that the y3 compartment is important in discriminative training, they would need to show no defects in single-odor conditioning as in Figure 6A.

As above, upon consideration of the reviewers’ comments and the overall theme/flow of the manuscript, we have removed the discriminative training experiment from the manuscript. We intend to follow up on this result more thoroughly in a future study. For now, we focus on the conclusion that the γ3 compartment exhibits alterations in ACh release, and that its downstream MBONs are required for appetitive conditioning.

- Selection of ROIs for different MBONs in Figure 5 need stronger rationale. Why do the authors choose neurites for some compartments and not for others? I guess there is transformation of calcium signals in different parts of MBONs.

We selected ROIs consistently with previous studies (e.g., Berry et al., 2018; Jacob and Waddell, 2020; Zhao et al., 2018), which allowed us to reliably image the Ca^2+^ responses in these neurons. This is now noted in the Methods.

- Describe how the authors defined compartments in the g-KCs?

Based on the criteria of Aso et al., 2014 (anatomical landmarks; see also Figure 1B).

- I'm not totally sure how the results in Figure 6 support "temporal comparison (L275)". Either elaborate this conclusion more or provide an alternative interpretation.

As noted above, we removed this experiment. It deserves a more thorough treatment in a follow‐up study.

- The quantified compartment for Figure 4A-D should be clearly stated in the text and legend, instead of simply showing y2 in 4E and let us guess.

Indeed, this oversight has been corrected – the caption now indicates the compartment.

- Is "post-conditioning odor contrast" appropriate for explaining itpr effects given the CS+ depression is seemingly exaggerated (Figure 4)?

It wasn’t – the time series traces are plotted as mean values, which are strongly influenced by large values (i.e., outliers). The CS+ was significantly different in both controls and IP_3_R RNAi groups (and the IP_3_R RNAi p value was in fact less significant). The most important data are the ΔF/F values, not the mean time series traces. The main effect is the presence of adaptation in the CS‐ group and the odor‐only groups when knocking down IP_3_R/itpr. We reworked the figure to include more data and focus on the important points (in this case, the pre and post ΔF/F values across all conditions).

Nonetheless, this raises an important related point ‐ the previous version of the manuscript left the role of IP_3_R/itpr somewhat unclear. To further bolster this aspect of the study, we carried out additional experiments and clarified our interpretations. Loss of the IP_3_R results in a reduction in adaptation to odors, which was noted in the initial manuscript as a reduction in odor‐evoked responses at the “post” time point across both experimental groups (CS+ and CS‐) and odor‐only controls. To explicitly test whether IP_3_R knockdown increased the rate of olfactory adaptation, we tested a second protocol: presenting and imaging the odor‐evoked ACh release from KCs in response to a 1‐s odor pulse, delivered once per minute for 10 minutes. This protocol revealed increased adaptation in the IP_3_R knockdown group relative to (uninduced) controls (Figure 4 K,L, Figure 4 – Supplement 2), demonstrating that IP_3_R is required for maintenance of olfactory responses over time.

- L256 Sentence and/or figure citation are somewhat inconsistent.

The reviewer is correct (the figure was cited erroneously). To address other concerns, this section has now been removed, as noted above.

[Editors’ note: what follows is the authors’ response to the second round of review.]

Reviewer #2:I've read the revised manuscript, and found it much improved. However, some of my concerns were not addressed. The authors need to clarify the following points before publication.1. The odor-only control accounts for the effect of non-associative olfactory adaptation. But only that. The authors should be aware that other types of non-associative plasticity taking place during learning; e.g. reduced odor acuity by the previous exposure to electric shock (Preat, J Neurosci, 1998). Thus, inclusion of the US-only and/or unpaired training controls is mandatory to formally determine the absolute plasticity effect (e.g. potentiation of CS+ or depression CS- are indistinguishable; Figures 1L and 2K). Otherwise, they should moderate these claims, and focus only on the relative effects (i.e. "CS+:CS-" in the same figures)..

We have clarified and moderated the interpretation of the findings, noting that the comparison normalizes for non‐associative olfactory adaptation, as the reviewer notes. This is done in multiple places, such as on page 5, which now reads: “The Δ(post/pre) of the CS+ and CS‐ was compared to determine how each changed relative to the other, and then each was compared to its respective odor‐only control to quantify whether it was potentiated or depressed, accounting for any nonassociative olfactory adaptation (Figure 1 G‐K, Figure 1 – Supplement 2).”

2. As demonstrated by the authors, Cac-KD reduces only ~29% of total Cac RNA level, suggesting that large body of Cac expression is unaffected. It is quite questionable that Cac is not required for learning-induced depression of Ach release (P10, L234) based on this partial down-regulation.

This is a valid point; we adjusted the language accordingly. It now reads: “Overall, these data suggest that moderate knockdown of Cac does not affect learning‐induced depression of ACh release (in contrast to potentiation).”

3. Somehow related to the previous point. Is it reasonable to conclude that Cac is required for learning-dependent potentiation? There is potentiation of CS- odor response in the γ 5 compartment after appetitive learning (Figure 3- sup1E)?

There is no statistically significant difference between the CS‐ and the odor‐only control. Therefore, while we are intrigued by the quantitative increase in the CS‐ response in this group (as the reviewer is), we are unable to ascribe it to an associative effect of conditioning.